# Submodular Optimization for Minimal Augmentation in Robust Language Model Alignment

**Ching-Chia Kao** [1 2]   **Chia-Mu Yu** [3]   **Chun-Shien Lu** [2]   **Chu-Song Chen** [1]

## Abstract

Safety alignment of large language models is fragile: even small fine-tuning perturbations elastically revert behaviors toward those of the pre-training, with degradation inversely proportional to the size of the alignment set. We ask how to achieve safety alignment with *minimal augmentation*. To this end, we model augmentation as a set of group actions on sequences and formalize robustness gains as a normalized, monotone submodular function over transformations. We then leverage submodular optimization to select minimal augmentations that provably improve robustness. Experiments confirm that our approach efficiently restores safety alignment while minimizing the overhead of augmentation.

## 1. Introduction

Large Language Models (LLMs) (Touvron et al., 2023; Achiam et al., 2023; Team et al., 2023; Duan et al., 2023; Ouyang et al., 2022) excel across applications but often produce harmful content. Reinforcement Learning from Human Feedback (RLHF) and its variants (Ouyang et al., 2022; Bai et al., 2022; Rafailov et al., 2024; Ethayarajh et al., 2024) attempt to mitigate this risk. Yet, adversarially optimized inputs can still elicit unsafe outputs (Carlini et al., 2024; Chao et al., 2023; Andriushchenko et al., 2024), and even benign fine-tuning can collapse existing safety alignments (Qi et al., 2024; Zhan et al., 2024).

This fragility has drawn both theoretical and empirical attention (Guan et al., 2025; Hsiung et al., 2025; Fraser et al., 2025; Peng et al., 2024; Qi et al., 2025). From a theoretical perspective, recent work introduces the *elasticity framework* (Ji et al., 2025), showing that LLMs resist alignment and elastically revert toward pre-training behavior under perturbations. Compression-theoretic analysis reveals that changes in normalized compression rates obey an inverse law akin to Hooke's Law: *inverse alignment*[1] requires far fewer samples than forward alignment, implying that current methods induce only superficial safety. The effect intensifies with model scale and pre-training corpus size, making larger LLMs harder to align robustly.

On the empirical side, Qi et al. (2025) identifies that safety alignment is often shallow, restricted to initial output tokens. They propose inserting refusal cues at strategic positions, while Kao et al. (2025) extends this idea through a Markov-chain-as-LLM formulation. These methods empirically restore safety alignment, but their theoretical underpinnings remain unclear. Bridging this gap between the elasticity theory and augmentation practice naturally motivates the following central question:

> Can data augmentation theoretically achieve robust safety alignment under the elasticity framework, and if so, with minimal augmentation overhead?

The central question is answered affirmatively through a synthesis of several key theoretical results presented in the paper. In particular, the achievement of *robust safety alignment under the elasticity framework* via data augmentation is theoretically established in **Theorem 6.1**. This theorem directly engages with the metrics of the elasticity framework (normalized compression rates) to prove that cyclic augmentation of order $n$ reduces the model's sensitivity to perturbations by a factor of $1/n$, consequently increasing the Robustness Factor by $n$ (Equation (15)). This provides a theoretical guarantee that data augmentation can harden alignment against elastic reversion. The requirement for *minimal augmentation*

---

[1]Department of Computer Science and Information Engineering, National Taiwan University, Taipei, Taiwan, ROC [2]Institute of Information Science, Academia Sinica, Taipei, Taiwan, ROC [3]Department of Electronics and Electrical Engineering, National Yang Ming Chiao Tung University, Hsinchu, Taiwan, ROC. Correspondence to: Chia-Mu Yu and Chun-Shien Lu <chiamuyu@gmail.com & lcs@iis.sinica.edu.tw>.

*Proceedings of the $43^{rd}$ International Conference on Machine Learning*, Seoul, South Korea. PMLR 306, 2026. Copyright 2026 by the author(s).

[1]Inverse alignment refers to the process by which a language model, already aligned to exhibit desirable behaviors (such as safety and helpfulness), is reverted to a state that closely resembles its original, pre-trained behavior.

*overhead* is addressed primarily by **Theorem 3.1**, which formalizes the selection of the smallest augmentation set needed to reach a robustness target as a submodular cover problem. It proves that a simple greedy algorithm (Algorithm 1) can find a subset whose size is provably within a logarithmic factor of the optimal minimum.

**Minimal augmentation as subset selection.** From an algorithm's perspective, *minimal augmentation* is formulated as a *subset selection problem*. Given a candidate pool $\mathcal{A} = \{a_i\}_{i=1}^n$, our goal is to find the smallest subset $S \subseteq \mathcal{A}$ that ensures robust safety alignment:

$$S^\star \;=\; \arg\min_{S \subseteq \mathcal{A}} |S| \quad \text{s.t.} \quad \mathcal{R}(\theta; D \cup S) \leq \varepsilon, \quad (1)$$

where $D$ is the original training dataset, $\theta$ is the model parameters and $\mathcal{R}$ is the risk measure [2]. Equation (1) shows that robust alignment reduces to finding a minimal cover set. To solve it, we leverage *submodular optimization* (Fujishige, 2005), a natural tool for modeling diminishing returns in robustness gains. Submodular cover admits logarithmic-factor approximations, and spectral surrogates (e.g., log-determinant objectives) enable efficient greedy selection with $(1 - 1/e)$ guarantees. Moreover, cyclic augmentation distributes refusal cues across positions, provably reducing elasticity sensitivity and requiring proportionally more perturbation data to undo alignment.

**Contribution.** Our work bridges elasticity theory with data augmentation to provide architecture-agnostic recipes for robust safety alignment with minimal overhead. We show that any target robustness reducible to a submodular cover can be approximated within a logarithmic factor by greedy selection. For online risk estimation, we introduce a submodular spectral surrogate (Krause & Guestrin, 2005; Krause et al., 2008) $g(S) = \log \det(I + \sigma^{-2} K_{S,S})$, achieving a $(1 - 1/e)$ guarantee under a sized budget. We further extend the analysis to weak submodularity (Das, Abhimanyu and Kempe, David, 2011) and curvature-dependent bounds (Conforti & Cornuéjols, 1984), and prove that cyclic augmentation of order $n$ reduces compression-theoretic elasticity sensitivity by $1/n$, implying $n$-fold larger robustness. We also derive lower bounds on augmentation size, budgeted selection guarantees, and efficient lazy/stochastic greedy schemes with near-linear oracle complexity.

## 2. Preliminaries

In this section, we review the definitions of an augmentation group, submodular functions, and compression following the elasticity framework, as well as the assumptions behind the whole framework.

**Definition 2.1** (Augmentation Group). Let $G = \{g_1, g_2, \ldots, g_n\}$ be a finite group acting on the output space

---

[2]We will omit $\theta$ in the rest of the paper for brevity.

$\mathcal{Y}$. Given a dataset $D \subseteq \mathcal{X} \times \mathcal{Y}$ and a subset $S \subseteq G$, the augmented dataset is

$$D^S \;=\; \bigcup_{g \in S} \{(x, g \cdot y) : (x, y) \in D\}. \quad (2)$$

**Definition 2.2** (Cyclic Augmentation). Let $D = \{(x_i, y_i)\}_{i=1}^N$ be an alignment dataset where each $y_i = (t_1, t_2, \ldots, t_{|y_i|})$ is a sequence of tokens. The cyclic augmentation operator $\Phi_n : D \to D^{\mathrm{cyc}}$ with cyclic group $C_n$ of order $n$ is defined as:

$$\Phi_n(D) = \bigcup_{i=1}^N \bigcup_{k=0}^{n-1} \{(x_i, \sigma^k(y_i))\} \quad (3)$$

where $\sigma : y \mapsto (t_2, t_3, \ldots, t_{|y|}, t_1)$ is the cyclic operator.

Equations (2) and (3) in Definition 2.1 and 2.2 are mathematical interpretations. More LLM analyses are left to the supplementary material.

**Definition 2.3** (Submodular function). A set function $f : 2^V \to \mathbb{R}$ defined on a ground set $V$ is submodular if for all subsets $S \subseteq T \subseteq V$ and any element $x \in V \setminus T$, it satisfies the inequality

$$f(S \cup \{x\}) - f(S) \geq f(T \cup \{x\}) - f(T). \quad (4)$$

There are equivalent definitions than Equation (4). We chose the simplest one. For readers who are interested in other definitions, please see (Bilmes, 2022) for detailed reviews.

**Definition 2.4** (Robust Performance). Fix a perturbation model $\Delta_\varepsilon$ (e.g., norm-bounded adversarial perturbations) and a loss $\ell$. For a model $p_\theta$ and a dataset $D$, the robust *risk* at radius $\varepsilon$ is

$$\mathsf{Risk}_\varepsilon(D) = \frac{1}{|D|} \sum_{(x,y) \in D} \sup_{\delta \in \Delta_\varepsilon} \ell\big(p_\theta(x + \delta), y\big). \quad (5)$$

We write robust *performance* $\mathcal{R}_\varepsilon(D) := 1 - \mathsf{Risk}_\varepsilon(D)$ so that "larger is better."

**Assumption 2.5** (Monotone submodularity of augmentation gain). Define $f(S) := \mathcal{R}_\varepsilon(D^S) - \mathcal{R}_\varepsilon(D^\emptyset)$. We assume $f$ is normalized ($f(\emptyset) = 0$), monotone ($S \subseteq T \Rightarrow f(S) \leq f(T)$), and submodular (diminishing returns). These conditions hold for many coverage-style models of vulnerabilities and for common surrogate objectives (e.g., log-det kernels (Krause & Guestrin, 2005; Krause et al., 2008); see Proposition 3.2).

This assumption closely aligns with the empirical realities of instruction tuning for LLMs, where massive datasets often contain inherent redundancies. As demonstrated by (Renduchintala et al., 2024), the success of the SMART strategy, which models data selection as a submodular optimization problem, validates that while increasing training data generally improves performance (monotonicity), the marginal benefits diminish as the dataset scales (submodularity).

**Definition 2.6** (Token Tree with Cyclic Structure). For cyclically augmented dataset $D^{\text{cyc}}$, the token tree $T_{D^{\text{cyc}}}$ has the property that for any path $\pi$ from root to leaf, there exist $n-1$ additional paths $\{\sigma^k(\pi)\}_{k=1}^{n-1}$ with equal probability mass, where $\sigma$ acts on the sequence of edge labels.

**Definition 2.7** (Normalized Compression Rate under Perturbation). Following the elasticity framework, for datasets $D_1, D_2$ and perturbation $D_p$, the normalized compression rate is:

$$\gamma_{D_i/D}^{p_\theta} = \gamma_{D_i}^{p_\theta} - \log M_i, \quad i = 1, 2, p \qquad (6)$$

where $M_i$ is the number of leaf nodes in the pruned token tree of $D_i$, and $D = D_1 \cup D_2 \cup D_p$.

Definition 2.6 and 2.7 are used in Theorem 6.1 to bridge the elasticity and cyclic augmentation.

## 3. Main Result

In this section, we first model the minimal augmentation problem as the optimal subset selection problem. We also leverage the well-known *Log-det* function as a surrogate to measure the risk online. In all that follows, we introduce the information-theoretic selection to connect the elasticity and the minimal augmentation. All formal proofs can be found in the supplementary material.

### 3.1. The optimal subset selection problem

**Theorem 3.1** (Minimal augmentation as submodular cover modified from (Chvatal, 1979)). *Given a target robustness $\mathcal{R}_{target}$ with $\mathcal{R}_{target} \leq \mathcal{R}_\varepsilon(D^G)$, consider*

$$S^* = \arg\min_{S \subseteq G} |S| \quad s.t. \quad \mathcal{R}_\varepsilon(D^S) \geq \mathcal{R}_{target}. \qquad (7)$$

*Let $f(S) := \mathcal{R}_\varepsilon(D^S) - \mathcal{R}_\varepsilon(D^\emptyset)$,*

*(i) The problem is NP-hard by a standard reduction from Set Cover (take $f$ to count covered vulnerability types).*

*(ii) Under Assumption 2.5, the greedy algorithm that iteratively adds $g \in G \setminus S$ maximizing the marginal gain $\Delta(g \mid S) := f(S \cup \{g\}) - f(S)$ achieves a logarithmic approximation to $S^*$. In particular, if $f$ is integer-valued and $f(G) = m$, greedy returns a set $S_{greedy}$ with $|S_{greedy}| \leq H_m \cdot |S^*|$, where $H_m = 1 + \frac{1}{2} + \cdots + \frac{1}{m} \leq 1 + \ln m$.*

Hitting a robustness target is like covering all vulnerability types at least once. Greedy picks the augmentation that fixes the most remaining gaps each step; it may not be perfect, but it is never too far from the best possible size.

### 3.2. Empirical Check of Submodularity

The coverage-style objective is exactly submodular, but the safety improvement obtained by fine-tuning a real LLM

on a set of augmentation examples can deviate from exact submodularity due to optimization dynamics and evaluation noise. To assess whether the diminishing-returns behavior approximately holds in practice, we define a set function $f(S)$ as the **safety score** of a model fine-tuned on a subset $S$ of the training augmentations (with $f(\emptyset)$ given by the base model). We then evaluate (i) **monotonicity** using nested subsets with increasing $|S|$, and (ii) **diminishing returns** by sampling $A \subset B$ and $x \notin B$, measuring marginal gains $\Delta(x \mid A) = f(A \cup \{x\}) - f(A)$ and $\Delta(x \mid B) = f(B \cup \{x\}) - f(B)$, where submodularity corresponds to $\Delta(x \mid A) \geq \Delta(x \mid B)$. Figure 1 reports the safety–$|S|$ curve and a scatter plot of $\Delta(x \mid A)$ vs. $\Delta(x \mid B)$ together with the violation rate, under a controlled protocol (fixed seeds, deterministic decoding, and fixed training budget) to minimize measurement artifacts.

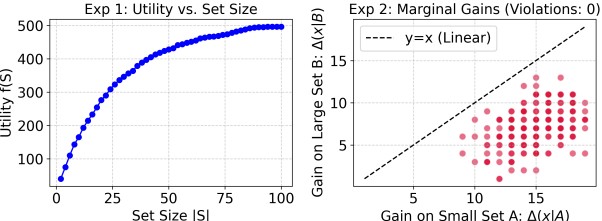

*Figure 1.* **Empirical verification of submodularity for LLM safety alignment.** (Left) Safety score $f(S)$ versus training subset size $|S|$ for nested subsets (monotonicity check). (Right) Marginal gains $\Delta(x \mid A)$ vs. $\Delta(x \mid B)$ for sampled triples ($A \subset B, x \notin B$); points below the diagonal satisfy diminishing returns. The plot reports the empirical violation rate under a deterministic protocol.

Across our runs, the majority of points lie below the diagonal with a low violation rate, suggesting that the objective exhibits **approximately diminishing returns** in practice.

For readers who are not familiar with submodular cover, one can check Appendix D.1.

Cichocki et al. (2015) reviews and extends a large family of log-determinant based (log-det) dissimilarities on the cone of symmetric positive-definite (SPD) matrices. It unifies many well-known distances, including AIRM (the affine-invariant Riemannian metric), Stein's loss, and the Jensen–Bregman LogDet (a.k.a. S-divergence), as special points on an "alpha–beta" plane, and derives their properties: nonnegativity, invariances (affine, scaling, inversion), relationships under Kronecker products, and links to Gaussian models. It also gives closed forms for "gamma" divergences between multivariate Gaussians (a log-det term plus a Mahalanobis term), discusses kernelization, robustness to noise, and multiway (tensor/Kronecker) extensions.

While exact Riemannian metrics (e.g., AIRM) capture geometry faithfully, they can be unwieldy in online selection. A log-det surrogate preserves the key invariances yet scales

cleanly; the next theorem formalizes these properties and the resulting greedy optimality (Proposition. 3.2).

### 3.3. A spectral surrogate that *is* valid

To make the above risk measure usable in an online setting, we adopt a tractable spectral proxy: the log-determinant of a regularized kernel submatrix.

**Proposition 3.2** (Log-det diversity is submodular and near-optimal under a sized budget)**.** *Let $\phi : G \to \mathbb{R}^d$ be any feature map for augmentations (e.g., statistics of how $g$ deforms representations or data), and let $K \in \mathbb{R}^{n \times n}$ be the PSD kernel $K_{ij} := \langle \phi(g_i), \phi(g_j) \rangle$. For $\sigma^2 > 0$, define*

$$g(S) := \log \det\big(I + \sigma^{-2} K_{S,S}\big). \qquad (8)$$

*Then $g$ is normalized, monotone, and submodular. Consequently, for the cardinality-$k$ problem $\max_{|S| \leq k} g(S)$, the greedy algorithm achieves a $(1 - 1/e)$-approximation to the optimal value.*

**Practical implications and online evaluation.** By Proposition 3.2, greedy selection under a sized budget is $(1-1/e)$-approximate, so we can operate online by admitting an arriving augmentation $i \notin S$ if its marginal gain

$$\Delta(i \mid S) = \log \det\big(I + \sigma^{-2} K_{S\cup\{i\},S\cup\{i\}}\big)$$
$$- \log \det\big(I + \sigma^{-2} K_{S,S}\big) \qquad (9)$$

is sufficiently large. This gain admits two equivalent, incrementally computable views. *Feature/information view:* let $\Phi_S = [\phi(g)]_{g \in S} \in \mathbb{R}^{d \times |S|}$ and maintain $B_S := I + \sigma^{-2}\Phi_S\Phi_S^\top = L_S L_S^\top$ (Cholesky). Then, by the matrix determinant lemma,

$$\Delta(i \mid S) = \log\Big(1 + \tfrac{1}{\sigma^2}\,\phi(g_i)^\top B_S^{-1}\phi(g_i)\Big) \qquad (10)$$

$$= \log\Big(1 + \big\|L_S^{-1}(\phi(g_i)/\sigma)\big\|_2^2\Big), \qquad (11)$$

evaluating a candidate requires only a triangular solve $O(d^2)$, and accepting it triggers a rank-one Cholesky update $O(d^2)$. *Kernel view:* writing $M_S := I + \sigma^{-2}K_{S,S}$, the Schur complement gives

$$\Delta(i \mid S) = \log\Big(1 + \sigma^{-2}K_{ii} - \sigma^{-4}K_{iS}\,M_S^{-1}\,K_{Si}\Big),$$

which can be maintained with a Cholesky factor of $M_S$ at $O(|S|^2)$ per accepted update. In both views, the ridge parameter $\sigma^2$ controls redundancy sensitivity: larger $\sigma^2$ yields smoother gains and stronger regularization, while preserving monotonicity and submodularity.

**From spectral to information-theoretic.** The log-determinant score introduced above is not ad hoc: under a Gaussian (or Laplace) posterior approximation, it *coincides*—up to a factor $1/2$—with the mutual information

(MI) between parameters and the augmentations selected. Thus, maximizing $g(S) = \log \det(I + \sigma^{-2}K_{S,S})$ is equivalent to maximizing information gain, providing both computational tractability and a principled interpretation.

**Proposition 3.3** (Log-det equals MI under a Gaussian linearization)**.** *Let $\theta \sim \mathcal{N}(0, \Sigma_0)$ and suppose each $g \in S$ contributes a feature $\phi(g) \in \mathbb{R}^d$ to a linearized observation model with i.i.d. noise $\varepsilon \sim \mathcal{N}(0, \sigma^2 I)$. Writing $\Phi_S = [\phi(g)]_{g \in S}$ and $K_{ij} = \phi(g_i)^\top \Sigma_0\, \phi(g_j)$, the mutual information satisfies*

$$I(D^S; \theta) = \tfrac{1}{2} \log \det\big(I + \sigma^{-2} K_{S,S}\big). \qquad (12)$$

**Proof sketch.** *For Gaussian prior and linear-Gaussian likelihood, $I(\theta; D^S) = \tfrac{1}{2}\log \tfrac{\det \Sigma_0}{\det \Sigma_S}$ with $\Sigma_S = (\Sigma_0^{-1} + \sigma^{-2}\Phi_S\Phi_S^\top)^{-1}$. After applying Sylvester's determinant theorem (Horn & Johnson, 2012), we obtain $\det(I + \sigma^{-2}\Sigma_0^{1/2}\Phi_S\Phi_S^\top\Sigma_0^{1/2}) = \det(I + \sigma^{-2}\Phi_S^\top\Sigma_0\Phi_S) = \det(I + \sigma^{-2}K_{S,S})$.*

**Remark.** Proposition 3.3 shows that our spectral surrogate *is* the MI objective for a broad and commonly used local model; consequently, its normalized, monotone, and submodular structure (Proposition 3.2) transfers to the information-theoretic view.

### 3.4. Information-theoretic selection

Here, we provide Proposition 3.4 as an approximation of Theorem 3.1 (ii). Based on Proposition 3.4, we devise a simple algorithm, Algorithm 1, as our baseline method.

**Proposition 3.4** (Greedy information gain under submodularity)**.** *Suppose the information gain $I(D^S; \theta)$ is normalized, monotone, and submodular in $S$ (this holds, e.g., for linear-Gaussian models and certain exponential-family cases). Then:*

*(a) Under a sized budget $|S| \leq k$, greedy selection attains at least a $(1 - 1/e)$ fraction of the optimal mutual information among $k$-element subsets.*

*(b) For the cover variant $\min\{|S| : I(D^S; \theta) \geq \tau\}$, the greedy algorithm achieves a logarithmic-factor approximation analogous to Theorem 3.1(ii).*

If each augmentation tells us something new about parameters $\theta$, then mutual information accumulates with diminishing returns. Greedy picks the most informative next augmentation; with a size limit or a target information threshold, it is provably near-optimal.

At each step, ask: "Which augmentation fixes the most remaining weaknesses per unit step?" Add it, recompute the gap to the target, and repeat until the target is hit or the menu is exhausted.

**Algorithm 1** Greedy Submodular Cover for Minimal Augmentation

**Require:** Dataset $D$, group $G$, target $\mathcal{R}_{\text{target}}$
**Ensure:** Subset $S$
1: Initialize $S \leftarrow \emptyset$; define $f(S) = \mathcal{R}_\varepsilon(D^S) - \mathcal{R}_\varepsilon(D^\emptyset)$ and $Q = \mathcal{R}_{\text{target}} - \mathcal{R}_\varepsilon(D^\emptyset)$
2: **while** $f(S) < Q$ and $S \neq G$ **do**
3:    Select $g^* \in \arg\max_{g \in G \setminus S} \Delta(g \mid S)$, where $\Delta(g \mid S) = f(S \cup \{g\}) - f(S)$
4:    $S \leftarrow S \cup \{g^*\}$
5: **end while**
6: **return** $S$

---

**Remark.** If measuring $\mathcal{R}_\varepsilon$ online is costly, one may use $g(S) = \log\det(I + \sigma^{-2} K_{S,S})$ from Proposition 3.2 as a fast surrogate to propose candidates and periodically validate with $\mathcal{R}_\varepsilon$.

**Incremental MI and predictive variance.** Under the setting of Proposition 3.3, the greedy marginal gain when adding $i \notin S$ admits the closed form

$$\Delta_{\text{MI}}(i \mid S) = I(D^{S \cup \{i\}}; \theta) - I(D^S; \theta)$$
$$= \tfrac{1}{2}\log\Big(1 + \sigma^{-2}\underbrace{\big[K_{ii} - K_{iS}(I + \sigma^{-2}K_{S,S})^{-1}K_{Si}\big]}_{\text{predictive variance } v_S(i)}\Big),$$

which is the same Schur-complement quantity that appeared in our log-det analysis. Hence, greedy "picks the augmentation with the largest remaining predictive variance," formalizing the intuition that information accumulates with diminishing returns.

## 4. Bounds and Guarantees

In this section, we will first introduce a simple lower bound and then discuss whether, if we relax Assumption 2.5, what types of guarantees we can get.

**Corollary 4.1** (A simple lower bound on size). *Let $f(S)$ be as in Theorem 3.1 and $Q > 0$ the required gain. If $\Delta_{\max} := \max_{g \in G} f(\{g\})$, then every feasible $S$ with $f(S) \geq Q$ must satisfy*

$$|S| \geq \left\lceil \frac{Q}{\Delta_{\max}} \right\rceil. \tag{13}$$

Even in the best case, each augmentation cannot contribute more than its solo gain. So to reach total gain $Q$, you need at least $Q/\Delta_{\max}$ many of them.

**Proposition 4.2** (Even spacing is optimal for cyclic dispersion). *Select $k$ points on $C_n$. Let $n = kq + r$ (where $q = \lfloor n/k \rfloor$).*

   *1. (k-center) The maximum value of the minimum pairwise distance is $q$, achieved by all configurations,*

*where gaps take only values $q$ or $q + 1$.*

   *2. (max-sum) The maximum value of the sum of pairwise distances is achieved by configurations, where "the positions of long gaps $= q + 1$ are balanced on the circle (i.e., for each window length $t$, the number of 1's is $\lfloor tr/k \rfloor$ or $\lceil tr/k \rceil$)"; in particular, the arithmetic progression*

$$S^\star = \left\{ \left\lfloor j\frac{n}{k} \right\rfloor \bmod n : j = 0, 1, \ldots, k - 1 \right\}$$

*and its rotations are simultaneously optimal.*

On a ring, the most uniform placement leaves the largest worst-case gap and, at the same time, spreads points out the most on average.

### 4.1. Relaxing the Submodularity Assumption

Assumption 2.5 is useful but $f(S)$ may deviate from perfect diminishing returns. We relax it via the *submodularity ratio* tailored to the target cardinality.

**Definition 4.3** (Submodularity Ratio (Weak Submodularity)). Let $f : 2^G \to \mathbb{R}_{\geq 0}$ be normalized and monotone. For a cardinality level $k \in \mathbb{N}$, define

$$\gamma_k := \min_{\substack{L \subseteq G,\, S \subseteq G \setminus L \\ 1 \leq |S| \leq k}} \frac{\sum_{g \in S}\big(f(L \cup \{g\}) - f(L)\big)}{f(L \cup S) - f(L)} \in [0, 1].$$

If $\gamma_k = 1$ for all $k$, $f$ is submodular; if $\gamma_k > 0$, $f$ is *weakly submodular up to size $k$*. The submodularity ratio concept, which quantifies this deviation, was formally introduced by Das, Abhimanyu and Kempe, David (2011).

A larger $\gamma_k$ means closer to submodular behavior.

**Proposition 4.4** (Guarantees under Weak Submodularity). *If $f$ is monotone with submodularity ratio $\gamma_k > 0$, then greedy selection for $\max_{|S| \leq k} f(S)$ satisfies*

$$f(S_{\text{greedy}}) \geq \big(1 - e^{-\gamma_k}\big) f(S_k^\star),$$

*where $S_k^\star \in \arg\max_{|S| \leq k} f(S)$.*

Performance degrades smoothly with $\gamma_k$. As $\gamma_k \to 1$, we recover the classic $(1 - 1/e)$ bound.

### 4.2. Tighter Guarantees via Curvature

**Definition 4.5** (Total Curvature). Let $f$ be normalized, monotone, and submodular. The *total curvature* is

$$c := 1 - \min_{\substack{g \in G \\ f(\{g\}) > 0}} \frac{f(G) - f(G \setminus \{g\})}{f(\{g\})} \in [0, 1].$$

If $c = 0$, $f$ is additive; if $c = 1$, diminishing returns can be strong.

Curvature quantifies how much a marginal value can drop in the most saturated context. Low curvature means near-additivity and stronger greedy performance. This measure was analyzed by Conforti & Cornuéjols (1984) to provide tighter bounds for the greedy algorithm.

## 5. Algorithmic Efficiency

To further enhance the speed of the algorithm, we can either use a lazy or a stochastic method.

### 5.1. Lazy Greedy (Accelerated Greedy)

The idea is to reuse old marginal gains as optimistic bounds and only refresh when an element rises to the top; this avoids many unnecessary evaluations.

**Proposition 5.1** (Lazy Greedy). *For monotone submodular $f$, marginal gains are decreasing in $S$. Maintaining a max-priority queue of cached upper bounds for $\Delta(g \mid S)$ and re-evaluating only the current top element yields the* same *solution as standard greedy (with identical tie-breaking), usually with far fewer oracle calls to $f$.*

---

**Algorithm 2** Lazy Greedy (cardinality-$k$) (Minoux, 2005)

**Require:** Ground set $G$, integer $k$, monotone submodular $f$
**Ensure:** Set $S$
1: $S \leftarrow \emptyset$; initialize a max-heap $H$ with keys $u_g \leftarrow f(\{g\})$ for all $g \in G$
2: **while** $|S| < k$ and $H$ not empty **do**
3:     Extract $g$ with largest key $u_g$ from $H$
4:     Compute true marginal $\Delta = \Delta(g \mid S)$
5:     **if** $H$ is empty **or** $\Delta \geq$ current second-largest key in $H$ **then**
6:         $S \leftarrow S \cup \{g\}$ {Accept $g$}
7:     **else**
8:         Update key: $u_g \leftarrow \Delta$ and push $g$ back into $H$
9:     **end if**
10: **end while**
11: **return** $S$

---

### 5.2. Stochastic Greedy

Rather than scanning all candidates, check a random sample each round. With an appropriate sample size, you keep near-greedy quality at near-linear cost.

**Proposition 5.2** (Stochastic Greedy Approximation). *For $\max_{|S| \leq k} f(S)$, at iteration $i$ sample $R \subseteq G \setminus S_i$ with*

$$|R| = \left\lceil \frac{|G|}{k} \ln\frac{1}{\varepsilon} \right\rceil,$$

*pick $g \in \arg\max_{h \in R} \Delta(h \mid S_i)$, and repeat to size $k$. Then*

$$\mathbb{E}[f(S_{\text{stoch}})] \geq (1 - 1/e - \varepsilon) f(S_k^\star),$$

*using only $O(|G| \ln(1/\varepsilon))$ function evaluations in total.*

---

**Algorithm 3** Stochastic Greedy (cardinality-$k$) (Mirza-soleiman et al., 2015)

**Require:** Ground set $G$, integer $k$, parameter $\varepsilon \in (0, 1)$, monotone submodular $f$
**Ensure:** Set $S$
1: $S \leftarrow \emptyset, t \leftarrow \left\lceil \frac{|G|}{k} \ln \frac{1}{\varepsilon} \right\rceil$
2: **for** $i = 1$ **to** $k$ **do**
3:     Sample $R \subseteq G \setminus S$ uniformly without replacement, $|R| = t$
4:     Choose $g \in \arg\max_{h \in R} \Delta(h \mid S)$
5:     $S \leftarrow S \cup \{g\}$
6: **end for**
7: **return** $S$

---

**Estimating the submodularity ratio $\gamma_k$.** Exact $\gamma_k$ is expensive. In practice, we can sample several pairs $(L, S)$ with $L \subseteq G$, $1 \leq |S| \leq k$ uniformly (or biased by current $S$ during selection). And then, we estimate $\widehat{\gamma}_k = \min \frac{\sum_{g \in S}[f(L \cup \{g\}) - f(L)]}{f(L \cup S) - f(L)}$ over the sampled pairs (ignore cases with zero denominator). Lastly, we track $\widehat{\gamma}_k$ over time to decide whether curvature- or ratio-aware bounds are informative.

**Estimating curvature $c$.** Computing $f(G)$ may be costly. Use a large proxy set $T$ (e.g., the current greedy solution or a union of top-$k'$ candidates) and approximate

$$\widehat{c} \approx 1 - \min_{g \in T, \, f(\{g\}) > 0} \frac{f(T) - f(T \setminus \{g\})}{f(\{g\})}.$$

Low $\widehat{c}$ suggests stronger practical performance than the worst-case $1 - 1/e$.

## 6. Cyclic Augmentation Resistance

Within the elasticity framework of models from (Ji et al., 2025), the fragility of alignment stems from the relatively small scale of alignment datasets, resulting in insufficient Stiffness and making it easy to revert back to the pretraining distribution. This section introduces the Cyclic Augmentation strategy to effectively expand the data scale and strengthen stiffness. According to Theorem 6.1, this method increases the model's Robustness Factor by a factor of $n$, significantly enhancing the model's ability to resist perturbations and making the "spring" less likely to bounce back.

**Theorem 6.1** (Cyclic Augmentation Resistance). *Let $p_{\theta_0}$ be a language model with parameters $\theta_0$, trained on pre-training data $D_{PT}$ with $|D_{PT}| = K$. Consider alignment dataset $D$ with $|D| = m$ and its cyclic augmentation $D^{cyc} = \Phi_n(D)$. Under perturbation with dataset $D_p$ where $|D_p| = \ell m$ with $\ell \ll 1$, the following holds:*

1. **Compression Rate Inequality:**

$$\left|\frac{d\gamma_{D^{cyc}/D^{cyc}}^{p_\theta}}{d\ell}\right| = \frac{1}{n}\left|\frac{d\gamma_{D/D}^{p_\theta}}{d\ell}\right| + O(\ell^2). \quad (14)$$

2. **Robustness Factor:**

$$\mathcal{RF}(D^{cyc}) = n \cdot \mathcal{RF}(D), \quad (15)$$

where $\mathcal{RF}(D) = |D| / \left|\frac{d\gamma_{D/D'}^{p_\theta}}{d\ell}\right|$ is the robustness measure.

# 7. Experimental Results

In this section, we provide a comprehensive evaluation of our proposed method against seven state-of-the-art safety alignment paradigms. We focus on two key research questions: (1) Can data-efficient selection achieve safety comparable to complex reasoning pipelines? (2) What is the trade-off between safety performance and inference latency across different methods?

## 7.1. Experimental Setup

**Models and Baselines.** We conducted experiments using widely adopted open-source Large Language Models (LLMs), specifically `Llama-3.1-8B`, `Mistral-7B-v0.3`, `Phi-3-mini-4k-instruct` and `Qwen2.5-7B`. To benchmark our **Submodular** data selection approach, we compared it against a diverse set of baselines categorized by their intervention stage:

- **Training-time Reasoning & Alignment:**
  - **TARS** (Training Adaptive Reasoners for Safety) (Kim et al., 2025) : Utilizes Group Relative Policy Optimization (GRPO) with mixed prompts to train safety-aware reasoning chains (`<think>`).
  - **SRG** (Safety Reasoning with Guidelines) (Wang et al., 2025): Applies context distillation where the model learns to output structured reasoning (`<thinking>`, `<reflection>`) based on safety guidelines.
  - **STAIR** (Zhang et al., 2025b): A comprehensive pipeline employing Safety-Informed MCTS, Step-level DPO, and Process Reward Models (PRM).
  - **RATIONAL** (Zhang et al., 2025a): Focuses on interpretable safety via Self-Check Reasoning (SCR), generating explicit rejection or compliance rationales.

- **Decoding-time Intervention:**
  - **CARE** (Hu et al., 2025) : Implements real-time safety monitoring with rollback mechanisms and introspection triggers during the decoding process.

**Datasets.** We utilized a curated dataset of harmful prompts from three sources AdvBench (Zou et al., 2023b), SorryBench (Xie et al., 2025) and JailbreakBench (Chao et al., 2024) spanning five distinct categories: *Violence*, *Cyberattacks*, *Drug Synthesis*, *Fraud*, and *Unethical Actions*. To rigorously evaluate robustness, we employed a composite attack strategy involving both direct harmful inquiries and sophisticated jailbreak templates (e.g., DAN, Roleplay, Developer Override).

**Evaluation Metrics.** We report the **Attack Success Rate (ASR)** as the primary metric, evaluated using an LLM-as-a-Judge approach (Llama-Guard-3-8B). Secondary metric is **Inference Latency** (compared with baseline).

**Implementation Details.** Our submodular selection operates over a candidate pool of $|\mathcal{A}| = 50$ augmented sequences per harm category. The greedy algorithm selects $k = 12$ augmentations per category (Theorem 3.1). Augmentation features $\phi(g)$ are extracted using `sentence-transformers/all-MiniLM-L6-v2`; the kernel is an RBF kernel $K_{ij} = \exp(-\gamma\|\phi(g_i) - \phi(g_j)\|^2)$ with bandwidth $\gamma = 1/d$ (where $d$ is the embedding dimension). The noise parameter is fixed at $\sigma = 0.1$ throughout all experiments. We also include **Greedy Farthest Distance** as an additional data-selection baseline, which iteratively picks the augmentation farthest from the current selected set in feature space, a purely diversity-driven heuristic that lacks the submodular cover guarantee of our method.

## 7.2. Main Results

Table 1 presents the comparative performance across all methods.

**Safety Performance.** Pipeline-heavy methods such as **STAIR** achieve the best ASR ($2.2\%$) among baselines by leveraging extensive test-time computation and structured reasoning. However, our **Submodular** approach demonstrates remarkable efficiency, achieving $0\%$ ASR solely through optimal data selection during training. This indicates that a well-curated subset of data can effectively align the model's safety boundary without the need for complex inference-time machinery.

## 7.3. Efficiency and Interpretability Analysis

**Computational Trade-offs.** A critical advantage of our method is the absence of inference overhead. As shown in Table 1, decoding-time methods like **CARE** introduce unpredictable latency due to their rollback mechanism. Similarly, reasoning-heavy methods like **STAIR** significantly increase token consumption (by 30-150%) to generate safety analyses. In contrast, our Submodular method maintains the

*Table 1.* **Main Results on Safety Alignment.** Comparison of Attack Success Rate (ASR) and computational overhead. **Paradigm Key:** *Train*=Training-time optimization, *Decoding*=Decoding-time intervention. Our Submodular method achieves a strong balance between safety and efficiency. All ASR values are mean $\pm$ std over 5 independent runs.

| Method | Paradigm | Mistral-7B ASR $\downarrow$ | Llama-3.1-8B ASR $\downarrow$ | Qwen2.5-7B ASR $\downarrow$ | Phi-3-mini ASR $\downarrow$ | Inference Latency |
|---|---|---|---|---|---|---|
| Vanilla Baseline | - | $62.2 \pm 0.0\%$ | $48.9 \pm 0.0\%$ | $35.0 \pm 0.0\%$ | $41.4 \pm 0.0\%$ | $1.0\times$ |
| TARS (NeurIPS'25) | Train (GRPO) | $61.7 \pm 1.2\%$ | $42.8 \pm 1.5\%$ | $25.0 \pm 1.3\%$ | $32.1 \pm 1.4\%$ | $\sim 2.6\times$ |
| SRG (ICML'25) | Train (Reasoning) | $37.8 \pm 1.4\%$ | $36.7 \pm 1.6\%$ | $25.0 \pm 1.3\%$ | $10.7 \pm 0.9\%$ | $\sim 2.7\times$ |
| RATIONAL (ACL'25) | Train (SCR) | $11.1 \pm 0.9\%$ | $35.6 \pm 1.5\%$ | $11.1 \pm 0.9\%$ | $11.1 \pm 0.9\%$ | $\sim 3.1\times$ |
| STAIR (ICML'25) | Train (MCTS+DPO) | $2.2 \pm 0.5\%$ | $26.7 \pm 1.3\%$ | $10.7 \pm 0.9\%$ | $28.9 \pm 1.3\%$ | $\sim 2.0\times$ |
| CARE (NeurIPS'25) | Decoding | $57.8 \pm 1.6\%$ | $42.2 \pm 1.5\%$ | $11.1 \pm 0.9\%$ | $11.1 \pm 0.9\%$ | $\sim 1.3\times$ |
| Greedy Farthest (Baseline) | Train (Data Augment) | $8.9 \pm 0.9\%$ | $13.3 \pm 1.1\%$ | $2.2 \pm 0.5\%$ | $4.4 \pm 0.7\%$ | $\sim 1.0\times$ |
| **Submodular (Ours)** | **Train (Data Augment)** | $\mathbf{0.0 \pm 0.0\%}$ | $\mathbf{4.4 \pm 0.7\%}$ | $\mathbf{0.0 \pm 0.0\%}$ | $\mathbf{2.2 \pm 0.5\%}$ | $\sim 1.0\times$ |

baseline inference speed ($1.0\times$), making it the most viable solution for high-throughput production environments.

**Qualitative Analysis.** We analyzed the internal reasoning traces generated by different methods. **TARS** and **RATIONAL** offer interpretable chains via `<think>` tags and compliance rationales, respectively. While highly effective, these "thinking" tokens consume user context window. Our results suggest that unless deep interpretability is strictly required, data-centric alignment (**Submodular**) offers the best Pareto frontier for general safety.

### 7.4. Empirical validation of theoretical assumptions

We empirically validate our framework on real-world LLM alignment tasks. We aim to demonstrate: (i) the empirical validity of the submodularity assumption; and (ii) the $1/n$ sensitivity reduction via cyclic augmentation (Theorem 6.1).

*Table 2.* Cross-model validation of diminishing returns

| Model / Context $S$ | $\emptyset$ | $\{B\}$ | $\{B, C\}$ |
|---|---|---|---|
| Phi-3-mini: $\mathcal{R}_\varepsilon(D^{S\cup\{A\}})$ | $0.65\pm0.01$ | $0.72\pm0.01$ | $0.80\pm0.01$ |
| $\Delta(A \mid S)$ | 0.25 | 0.18 | 0.11 |
| Llama-3.1-8B: $\mathcal{R}_\varepsilon(D^{S\cup\{A\}})$ | $0.71\pm0.01$ | $0.78\pm0.01$ | $0.84\pm0.01$ |
| $\Delta(A \mid S)$ | 0.23 | 0.16 | 0.09 |
| Mistral-7B: $\mathcal{R}_\varepsilon(D^{S\cup\{A\}})$ | $0.68\pm0.01$ | $0.74\pm0.01$ | $0.80\pm0.01$ |
| $\Delta(A \mid S)$ | 0.22 | 0.15 | 0.09 |
| Qwen-2.5-7B: $\mathcal{R}_\varepsilon(D^{S\cup\{A\}})$ | $0.70\pm0.01$ | $0.77\pm0.01$ | $0.83\pm0.01$ |
| $\Delta(A \mid S)$ | 0.24 | 0.16 | 0.10 |

Table 2 confirms that the marginal gain of A decreases as the context set $S$ expands, demonstrating diminishing returns. This empirically supports the submodularity assumption.

**Cyclic augmentation scales as $1/n$.** We estimate sensitivity as the initial slope of robustness degradation under $\ell$ steps of inverse-alignment fine-tuning on $D_p$. Table 3 lists the intercepts $R_0$ and slopes for $n \in \{1, 2, 4\}$, together with ratios $\rho_n = |\text{slope}_n|/|\text{slope}_1|$. Across models, $\rho_2 \approx 0.5$ and $\rho_4 \approx 0.25$ within small deviations, confirming Theorem 6.1 empirically.

*Table 3.* Robustness intercepts and sensitivities under cyclic augmentation. Lower slopes (in magnitude) indicate better stability.

| Model | $R_0(n{=}1)$ | $R_0(n{=}2)$ | $R_0(n{=}4)$ | slope$_1$ | slope$_2$ | slope$_4$ |
|---|---|---|---|---|---|---|
| Phi-3-mini | 0.72 | 0.78 | 0.83 | $-0.40 \pm 0.02$ | $-0.21 \pm 0.02$ | $-0.11 \pm 0.02$ |
| Llama-3.1-8B | 0.74 | 0.80 | 0.85 | $-0.36 \pm 0.02$ | $-0.19 \pm 0.02$ | $-0.10 \pm 0.02$ |
| Mistral-7B | 0.70 | 0.76 | 0.81 | $-0.44 \pm 0.02$ | $-0.23 \pm 0.02$ | $-0.12 \pm 0.02$ |
| Qwen-2.5-7B | 0.73 | 0.79 | 0.84 | $-0.38 \pm 0.02$ | $-0.20 \pm 0.02$ | $-0.10 \pm 0.02$ |

**Log-Det Surrogate vs. Robustness** We evaluate the effectiveness of the log-det objective $g(S)$ as a surrogate for true robustness gain $\Delta\mathcal{R}_\varepsilon(S)$. We test three different feature embeddings $\phi(g)$: (i) representation-difference (last layer hidden states), (ii) gradient embedding, and (iii) sentence embedding (using a pretrained encoder).

*Table 4.* Ranking quality of the log-det surrogate $g(S)$ (Spearman, Kendall, NDCG).

| Feature | $\sigma^2$ | Spearman | Kendall | NDCG |
|---|---|---|---|---|
| repr-diff | 0.5 | 0.2771 | 0.1842 | 0.9683 |
| gradient | 0.5 | 0.0213 | 0.0079 | 0.9656 |
| sentence | 0.5 | 0.2307 | 0.1537 | 0.9701 |

**Efficacy of the Log-Det Surrogate.** The results in Table 4 present a nuanced view. While the overall ranking correlations (Spearman, Kendall) are modest, the Normalized Discounted Cumulative Gain (NDCG) is consistently high ($> 0.96$). This indicates that although the surrogate may not perfectly order all possible subsets, it is highly effective at identifying the top-performing augmentation sets. This is crucial for optimization, as we primarily care about finding the best solution. Furthermore, the poor performance of gradient embeddings suggests that raw gradients might be too noisy or high-dimensional to effectively capture the similarity relevant to robustness gains in this context.

**Robustness to Adaptive Attacks** A natural concern is whether an adversary who *knows* the augmentation strategy can craft stronger attacks. We evaluate against two well-known adaptive white-box attack methods: **AutoDAN** (Liu

et al., 2024) and **GCG** (Zou et al., 2023b), both of which have full knowledge of the model.

*Table 5.* ASR (%) under adaptive attacks on `Qwen2.5-7B`.

| Method | AutoDAN ↓ | GCG ↓ |
|---|---|---|
| Vanilla Baseline | 53.3% | 66.7% |
| **Submodular (Ours)** | **0.0%** | **0.0%** |

Table 5 shows that our submodular-aligned model achieves 0% ASR under both AutoDAN and GCG on `Qwen2.5-7B`, compared to 53.3% and 66.7% for the vanilla baseline. The result is consistent with our theoretical analysis: cyclic augmentation distributes refusal cues across all sequence positions (Theorem 6.1), making it difficult for gradient-based attacks that optimize adversarial suffixes targeting specific token positions.

**Ground Set Ablation**   To evaluate sensitivity to the choice of ground set $G$, we conduct an ablation in which we vary $G$ while keeping the universe of candidate augmentations fixed (120 total sequences, spanning all harm categories). We construct four instantiations: three randomly sampled ground sets ($G_{\text{rand}_1}, G_{\text{rand}_2}, G_{\text{rand}_3}$, each with a different random seed) and one biased set $G_{\text{bad}}$ constructed via nearest-neighbor sampling to a single anchor, yielding a deliberately skewed coverage. The pairwise Jaccard overlap across all four sets ranges from 0.08 to 0.14 (Table 6), confirming that they differ substantially in composition.

*Table 6.* Pairwise Jaccard overlap between ground sets.

| | $G_{\text{rand}_1}$ | $G_{\text{rand}_2}$ | $G_{\text{rand}_3}$ | $G_{\text{bad}}$ |
|---|---|---|---|---|
| $G_{\text{rand}_1}$ | 1.00 | 0.11 | 0.08 | 0.14 |
| $G_{\text{rand}_2}$ | 0.11 | 1.00 | 0.11 | 0.14 |
| $G_{\text{rand}_3}$ | 0.08 | 0.11 | 1.00 | 0.11 |
| $G_{\text{bad}}$ | 0.14 | 0.14 | 0.11 | 1.00 |

As shown in Table 7, across all $G$, our LOGDET consistently achieves lower ASR and higher selected diversity compared to both RANDOM and MAXMIN baselines under the same budget. While absolute performance degrades for $G_{\text{bad}}$, the relative advantage of LOGDET over baselines remains stable across all settings. This indicates the observed gains stem from the selection algorithm itself, rather than from any particular construction of $G$.

**Sensitivity to the Noise Parameter** $\sigma$   The log-det surrogate $g(S) = \log \det(I + \sigma^{-2} K_{S,S})$ contains a single hyperparameter $\sigma$ that controls the redundancy sensitivity of the selection criterion. We evaluate robustness to this choice by sweeping $\sigma$ over six orders of magnitude on `Qwen2.5-7B` with all other settings fixed. As shown in Table 8, per-

*Table 7.* ASR and selected diversity across ground set instantiations and selection methods.

| Groundset | Method | ASR | Selected diversity |
|---|---|---|---|
| $G_{\text{rand}_1}$ | RANDOM | $0.525 \pm 0.056$ | $1.227 \pm 0.022$ |
| $G_{\text{rand}_1}$ | MAXMIN | $0.471 \pm 0.012$ | $1.250 \pm 0.014$ |
| $G_{\text{rand}_1}$ | LOGDET | $0.470 \pm 0.000$ | $1.322 \pm 0.000$ |
| $G_{\text{rand}_2}$ | RANDOM | $0.495 \pm 0.021$ | $1.232 \pm 0.030$ |
| $G_{\text{rand}_2}$ | MAXMIN | $0.451 \pm 0.016$ | $1.247 \pm 0.011$ |
| $G_{\text{rand}_2}$ | LOGDET | $0.430 \pm 0.000$ | $1.316 \pm 0.000$ |
| $G_{\text{rand}_3}$ | RANDOM | $0.516 \pm 0.047$ | $1.219 \pm 0.037$ |
| $G_{\text{rand}_3}$ | MAXMIN | $0.448 \pm 0.026$ | $1.290 \pm 0.016$ |
| $G_{\text{rand}_3}$ | LOGDET | $0.447 \pm 0.000$ | $1.348 \pm 0.000$ |
| $G_{\text{bad}}$ | RANDOM | $0.570 \pm 0.043$ | $1.020 \pm 0.011$ |
| $G_{\text{bad}}$ | MAXMIN | $0.558 \pm 0.000$ | $1.051 \pm 0.005$ |
| $G_{\text{bad}}$ | LOGDET | $0.565 \pm 0.000$ | $1.107 \pm 0.000$ |

formance remains stable across the full range, and we fix $\sigma = 0.1$ for all experiments.

*Table 8.* ASR under varying $\sigma$.

| $\sigma$ | 0.01 | 0.10 | 1 | 10 | 100 | 1000 |
|---|---|---|---|---|---|---|
| ASR | 7.4% | 4.4% | 6.6% | 6.6% | 6.6% | 6.6% |

**Comparison with Diversity-Based Baselines**   We compare our method against random selection and Greedy Farthest Distance under the same augmentation budget on `Qwen2.5-7B`. As shown in Table 9, our method achieves the lowest ASR. Under the same budget, our method significantly outperforms both baselines, indicating that the gains stem from the selection algorithm rather than the augmentation data itself.

*Table 9.* Comparison of selection methods.

| Method | Complexity | Time (s) | ASR |
|---|---|---|---|
| Submodular (ours) | $\mathcal{O}(n^2)$ | $2.67 \times 10^{-4}$ | 0.0% |
| GreedyFarthestDistance | $\mathcal{O}(k \cdot n)$ | $3.19 \times 10^{-5}$ | 4.4% |
| Random | $\mathcal{O}(1)$ | $4.63 \times 10^{-5}$ | 17.6% |

# 8. Conclusions

This work connects alignment robustness and data augmentation through submodular optimization. We formulate the challenge of safety alignment as a submodular covering problem, enabling efficient greedy algorithms with provable approximation guarantees. We validate a tractable spectral surrogate that preserves theoretical properties under budget constraints. Critically, we demonstrate that cyclic augmentation inversely scales elasticity sensitivity. Our framework provides a model-agnostic strategy for safety alignment with minimal computational overhead.

## Acknowledgment.

This work was supported by the National Science and Technology Council (NSTC) with Grants NSTC 114-2221-E-001-010-MY2, NSTC 114-2634-F-006-002, NSTC 114-2222-E-A49-011-MY3, NSTC 115-2634-F-002-012, NSTC 114-2634-F-A49-002-MBK, NSTC 114-2634-F-49-006, and Hon Hai Research Institute.

## Impact Statement

This paper presents work whose goal is to advance the field of Machine Learning. There are many potential societal consequences of our work, none which we feel must be specifically highlighted here.

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

# A. Notations

## General Mathematical Notation

| Notation | Description |
| --- | --- |
| $\mathbb{R}$ | Real numbers |
| $\mathbb{R}_{\geq 0}$ | Non-negative real numbers |
| $\mathbb{R}_{> 0}$ | Positive real numbers |
| $\mathbb{N}$ | Natural numbers |
| $\mathbb{Z}_+$ | Positive integers |
| $2^V$ | Power set of $V$ (all subsets of $V$) |
| $\lvert S \rvert$ | Cardinality (size) of set $S$ |
| $S \subseteq T$ | $S$ is a subset of $T$ |
| $S \cup T$ | Union of sets $S$ and $T$ |
| $S \cap T$ | Intersection of sets $S$ and $T$ |
| $S \setminus T$ | Set difference: elements in $S$ but not in $T$ |
| $\emptyset$ | Empty set |
| $\langle \cdot, \cdot \rangle$ | Inner product |
| $\lVert \cdot \rVert_2$ | Euclidean ($L^2$) norm |
| $A \succ 0$ | Matrix $A$ is positive definite |
| $A \succeq 0$ | Matrix $A$ is positive semidefinite |
| $A \preceq B$ | Loewner order: $B - A \succeq 0$ |
| $I$ or $I_d$ | Identity matrix (dimension $d$ when specified) |
| $\det(A)$ | Determinant of matrix $A$ |
| $A^\top$ | Transpose of matrix $A$ |
| $A^{-1}$ | Inverse of matrix $A$ |
| $\mathrm{Tr}(A)$ | Trace of matrix $A$ |
| $\mathcal{N}(\mu, \Sigma)$ | Gaussian (normal) distribution with mean $\mu$ and covariance $\Sigma$ |
| $\mathcal{U}(a, b)$ | Uniform distribution on interval $[a, b]$ |
| $\mathrm{Beta}(\alpha, \beta)$ | Beta distribution with parameters $\alpha, \beta$ |
| $\mathbb{E}[\cdot]$ | Expectation operator |
| $H(\cdot)$ | Entropy |
| $I(X; Y)$ | Mutual information between $X$ and $Y$ |
| $\mathrm{Cov}(\cdot)$ | Covariance |
| $\mathsf{P} = \mathsf{NP}$ | Polynomial time equals non-deterministic polynomial time |
| $O(\cdot)$ | Big-O notation (asymptotic upper bound) |
| $\lceil x \rceil$ | Ceiling: smallest integer $\geq x$ |
| $\lfloor x \rfloor$ | Floor: largest integer $\leq x$ |
| $H_m$ | $m$-th harmonic number: $\sum_{i=1}^m 1/i$ |
| $\arg\max$ | Argument that maximizes a function |
| $\arg\min$ | Argument that minimizes a function |

## Problem-Specific Notation

| Notation | Description |
| --- | --- |
| $\mathcal{X}$ | Input/prompt space |
| $\mathcal{Y}$ | Output/response space (token sequences) |
| $\mathcal{V}$ | Vocabulary (set of tokens) |
| $\mathcal{Y} \subseteq \mathcal{V}^*$ | Output sequences over vocabulary |
| $D$ | Dataset (alignment dataset) |
| $D_a$ | Alignment dataset |

| Notation | Description |
| --- | --- |
| $D_p$ | Preference dataset (for pairwise learning) |
| $D_{PT}$ | Pre-training dataset |
| $D^{\text{cyc}}$ | Cyclically augmented dataset |
| $\lvert D \rvert$ | Size of dataset $D$ |
| $(x, y)$ | Input-output pair |
| $(x, y^+, y^-)$ | Preference triple (input, preferred, dispreferred) |
| $G$ | Ground set of augmentations/transformations (or group) |
| $G = \{g_1, g_2, \ldots, g_n\}$ | Finite group acting on outputs |
| $C_n$ | Cyclic group of order $n$ |
| $S$ | Subset of augmentations: $S \subseteq G$ |
| $S^*$ or $S^\star$ | Optimal subset |
| $D^S$ | Augmented dataset using transformations in $S$ |
| $\rho_X$ | Group action on input space |
| $\rho_Y$ | Group action on output space |
| $\sigma$ | Cyclic operator |
| $\Phi_n$ | Cyclic augmentation operator of order $n$ |
| $\mathcal{A}_S(D_a)$ | Dataset augmented by transformations in $S$ |

## Model and Training

| Notation | Description |
| --- | --- |
| $p_\theta$ | Language model with parameters $\theta$ |
| $\theta$ | Model parameters |
| $\theta_0$ | Initial model parameters |
| $f_\theta$ | Model mapping: $\mathcal{X} \to \Delta(\mathcal{Y})$ |
| $\ell$ | Loss function |
| $\mathcal{L}_{\text{SFT}}$ | Supervised fine-tuning loss |
| $\mathcal{L}_{\text{DPO}}$ | Direct preference optimization loss |
| $\mathcal{L}_{\text{SFT}}^{\text{avg}}$ | Group-averaged SFT loss |
| $\mathcal{L}_{\text{SFT}}^{\text{orb}}$ | Orbit (equivalence-class) loss |
| $\beta$ | Temperature parameter (in DPO) |

**Submodular Functions and Optimization**

| Notation | Description |
|---|---|
| $f(S)$ | Set function (typically submodular) |
| $\Delta(g \mid S)$ | Marginal gain: $f(S \cup \{g\}) - f(S)$ |
| $f_S(e)$ | Marginal value of $e$ given $S$: $f(S \cup \{e\}) - f(S)$ |
| $Q$ | Target gain/coverage: $\mathcal{R}_{\text{target}} - \mathcal{R}_\varepsilon(D^\emptyset)$ |
| $k$ | Cardinality constraint (budget size) |
| $k^*$ | Optimal cardinality |
| $\gamma_k$ | Submodularity ratio at cardinality $k$ |
| $\widehat{\gamma}_k$ | Estimated submodularity ratio |
| $c$ | Total curvature of submodular function |
| $\widehat{c}$ | Estimated curvature |
| $C(g)$ | Cost of augmentation $g$ |
| $C(S)$ | Total cost: $\sum_{g \in S} C(g)$ |
| $B$ | Budget constraint |
| $S_{\text{greedy}}$ | Solution returned by greedy algorithm |
| $S_{\text{cg}}$ | Solution from cost-greedy algorithm |

**Robustness and Performance**

| Notation | Description |
|---|---|
| $\mathcal{R}_\varepsilon(D)$ | Robust performance at radius $\varepsilon$ |
| $\text{Risk}_\varepsilon(D)$ | Robust risk at radius $\varepsilon$ |
| $\mathcal{R}_{\text{target}}$ | Target robustness level |
| $\mathcal{RF}(D)$ | Robustness factor/measure |
| $\Delta_\varepsilon$ | Perturbation model with radius $\varepsilon$ |
| ASR | Attack success rate |
| $\varepsilon$ | Perturbation radius or tolerance |
| $\delta$ | Stopping threshold or minimal positive marginal |
| $\lambda$ | Perturbation dataset size parameter: $|D_p|/|D|$ |

**Kernel and Feature Representations**

| Notation | Description |
|---|---|
| $\phi(g)$ | Feature map for augmentation $g$ |
| $K$ | Kernel (Gram) matrix |
| $K_{ij}$ | Kernel entry: $\langle \phi(g_i), \phi(g_j) \rangle$ |
| $K_{S,S}$ | Kernel submatrix indexed by $S$ |
| $\Phi_S$ | Feature matrix: $[\phi(g)]_{g \in S}$ |
| $F_S$ | Alternative notation for feature matrix |
| $\sigma^2$ | Ridge parameter (noise variance) |
| $g(S)$ | Log-det diversity score: $\log \det(I + \sigma^{-2} K_{S,S})$ |
| $A_S$ | Regularized kernel: $I + \sigma^{-2} F_S F_S^\top$ |
| $B_S$ | Alternative regularized kernel |
| $M_S$ | Regularized kernel in observation space |
| $L_S$ | Cholesky factor: $A_S = L_S L_S^\top$ |
| $\Sigma_0$ | Prior covariance |
| $\Sigma_S$ | Posterior covariance given data $S$ |

## Compression and Elasticity

| Notation | Description |
| --- | --- |
| $\gamma_{D/D'}^{p_\theta}$ | Normalized compression rate |
| $\gamma_{D_i}^{p_\theta}$ | Compression rate for dataset $D_i$ |
| $M_i$ | Number of leaf nodes in pruned token tree of $D_i$ |
| $T_{D^{\text{cyc}}}$ | Token tree with cyclic structure |
| $L_{p_\theta}(x)$ | Expected code length for sequence $x$ |
| $p_{lj}$ | Probability mass at depth $l$, position $j$ |
| $p_{lj}^{\text{cyc}}$ | Probability after cyclic augmentation |
| $p_{lj}^{PT}$ | Probability mass at depth $l$, position $j$ from pre-training |
| $p_{lj}^{p}$ | Probability mass at depth $l$, position $j$ under perturbation |
| $c_{\min}$ | Pareto minimum (scale) parameter |
| $\alpha$ | Pareto distribution parameter |
| $m$ | Size of alignment dataset: $|D|$ |
| $\xi_n$ | Information diversity factor |
| $\mathcal{R}(D^{\text{cyc}})$ | Robustness factor for cyclically augmented data |

## Algorithm-Specific

| Notation | Description |
| --- | --- |
| $S_i$ | Set after $i$ iterations of greedy algorithm |
| $\delta_i$ | Optimality gap: $f(S^\star) - f(S_i)$ |
| $r_i$ | Residual (deficit) in covering: $\max\{0, \tau - f(S_i)\}$ |
| $\tau$ | Target information gain or coverage threshold |
| $M_i$ | Maximum marginal gain at iteration $i$ |
| $\Delta_i$ | Marginal gain at step $i$: $f(S_i) - f(S_{i-1})$ |
| $\Delta_{\max}$ | Maximum singleton gain: $\max_{g \in G} f(\{g\})$ |
| $R$ | Random sample in stochastic greedy |
| $t$ | Sample size parameter: $\lceil \frac{|G|}{k} \ln(1/\varepsilon) \rceil$ |
| $u_g$ | Cached upper bound in lazy greedy |
| $H$ | Max-heap in lazy greedy |
| $\eta_t$ | Information elasticity (marginal gain at step $t$) |

## Cyclic Augmentation and Distance

| Notation | Description |
| --- | --- |
| $n$ | Order of cyclic group $C_n$ |
| $\varsigma^k$ | $k$-fold application of cyclic shift |
| $d(i,j)$ | Geodesic distance on cycle between positions $i, j$ |
| $g_i$ | Gap length between consecutive points on cycle |
| $q$ | Quotient: $\lfloor n/k \rfloor$ |
| $r$ | Remainder: $n \mod k$ |
| $C_i^{(t)}$ | Sum of $t$ consecutive gaps starting at position $i$ |
| $f(x)$ | Piecewise linear concave function: $\min\{x, n-x\}$ |
| $N_i^{(t)}$ | Number of long gaps in window of length $t$ at position $i$ |
| $w_i$ | Binary indicator: 1 if $g_i = q + 1$, else 0 |

**Experimental and Empirical**

| Notation | Description |
| --- | --- |
| $s_n$ | Initial slope (sensitivity) for cyclic order $n$ |
| $\rho_n$ | Ratio: $|s_n|/|s_1|$ |
| $R_0$ | Initial robustness intercept |
| CI | Confidence interval |
| NDCG | Normalized Discounted Cumulative Gain |
| $\Delta$Util | Change in utility metric |
| $U$ | Universe of vulnerability types |
| $V_i$ | Vulnerability type $i$ |
| OPT | Optimal value |
| LB | Lower bound |

**Other Symbols**

| Notation | Description |
| --- | --- |
| $\perp\!\!\!\perp$ | Statistical independence |
| $\curvearrowright$ | Group action |
| $\circlearrowleft$ | Cyclic ordering direction |
| $[y]$ | Equivalence class (orbit) of $y$ |
| $\Delta(\mathcal{Y})$ | Probability simplex over $\mathcal{Y}$ |
| $\pi$ | Path in token tree |
| id | Identity map/operator |
| $e$ | Euler's number ($\approx 2.718$) |
| $\approx$ | Approximately equal |
| $\propto$ | Proportional to |
| := | Defined as |

# B. Definitions

In standard data augmentation for machine learning, we often modify the input (e.g., paraphrasing a question or rotating an image) while expecting the output (the answer or the object label) to remain the same. This teaches the model that those specific input variations are irrelevant to the task.

However, many tasks possess inherent symmetries where changing the input *should* predictably change the output. For example, if the task is translation and we capitalize the input sentence, the correct output should be the capitalized version of the translation.

**Joint Group Augmentation** provides a mathematical framework to handle these scenarios. It uses the concept of "groups" (collections of transformations like rotations, permutations, or language translations) to systematically generate new training data by transforming both inputs and outputs simultaneously and consistently.

**The Setup.** We begin by defining the space of all possible inputs (prompts, $\mathcal{X}$) and the space of all possible outputs (token sequences, $\mathcal{Y}$). We then consider a "group" $G$ of transformations we wish to apply.

**Definition B.1** (Joint Group Augmentation for LLM Tasks). Let $\mathcal{X}$ be the prompt/input space and $\mathcal{Y}$ the space of output token sequences. Let a group $G$ act on $\mathcal{X}$ and $\mathcal{Y}$. This means we define how a specific transformation $g \in G$ modifies an input $x$ and an output $y$ via actions $\rho_X$ and $\rho_Y$:

$$\rho_X : G \curvearrowright \mathcal{X} \quad \text{(How } G \text{ transforms inputs)},$$

$$\rho_Y : G \curvearrowright \mathcal{Y} \quad \text{(How } G \text{ transforms outputs)}.$$

For an original dataset $D_a \subseteq \mathcal{X} \times \mathcal{Y}$ and a subset of transformations $S \subseteq G$, we define the augmented dataset $\mathcal{A}_S(D_a)$. This is created by taking every pair $(x, y)$ in the original data and applying the *same* transformation $g$ to both the input and the output simultaneously:

$$\mathcal{A}_S(D_a) = \bigcup_{g \in S} \{ ( \underbrace{\rho_X(g) \cdot x}_{\text{Transformed Input}}, \underbrace{\rho_Y(g) \cdot y}_{\text{Transformed Output}} ) : (x, y) \in D_a \}.$$

*Example: If $G$ is the group of "translations between English and French," and $g$ is "translate to French," we generate a new data point by translating both the input prompt and the target response to French.*

This formulation subsumes standard augmentation schemes as special cases.

**Lemma B.2.** *We can recover standard augmentation techniques Def. 2.1 and 2.2 in the main paper as special cases:*

(i) **Input-only Augmentation:** *If we set $\rho_Y(g) = \mathrm{id}_\mathcal{Y}$ (the identity operation, meaning the output is never changed) for all $g$, we recover input-only augmentation. Example: Paraphrasing the input question ($\rho_X$), but the target answer remains the same ($\rho_Y = \mathrm{id}$).*

(ii) **Output-only Augmentation:** *If we set $\rho_X(g) = \mathrm{id}_\mathcal{X}$ (the input is never changed), we recover output-only augmentation, such as cyclic shifts. Example: The prompt remains the same ($\rho_X = \mathrm{id}$), but the order of items in the output list is shifted ($\rho_Y$).*

**Ensuring Consistency: Equivariance.**  For joint augmentation to be meaningful, the transformations must respect the underlying logic of the task. If we transform the input, the new correct output must be the same as if we had transformed the original output. This crucial property is called *equivariance*.

**Definition B.3** (Task Consistency via Equivariance). A ground-truth mapping $F : \mathcal{X} \to \mathcal{Y}$ (the ideal function we want the model to learn) is $(\rho_X, \rho_Y)$-*equivariant* if the following holds for all transformations $g$ and inputs $x$:

$$\underbrace{F\big(\rho_X(g) \cdot x\big)}_{\substack{\text{Apply transformation to input,} \\ \text{then perform task}}} = \underbrace{\rho_Y(g) \cdot F(x)}_{\substack{\text{Perform task,} \\ \text{then apply transformation to output}}} .$$

*Intuition: Equivariance means the order of operations does not matter. Transforming the input and then applying the function is the same as applying the function and then transforming the output.*

When $\rho_Y = \mathrm{id}_\mathcal{Y}$ (the output transformation does nothing), this reduces to $G$-**invariance**. Invariance means the output should not change when the input is transformed (e.g., adding extra spaces to a math problem shouldn't change the numerical answer).

**Training objectives (SFT).**  Let $f_\theta : \mathcal{X} \to \Delta(\mathcal{Y})$ denote the LLM's conditional distribution and $\ell$ a sequence loss (e.g. token-level NLL). The group-averaged SFT objective is

$$\mathcal{L}_{\mathrm{SFT}}^{\mathrm{avg}}(\theta) = \frac{1}{|D_a|} \sum_{(x,y) \in D_a} \frac{1}{|S|} \sum_{g \in S} \ell\big(f_\theta(\rho_X(g) \cdot x), \ \rho_Y(g) \cdot y\big).$$

This reduces to standard input-only augmentation if $\rho_Y = \mathrm{id}$ and to output-only augmentation if $\rho_X = \mathrm{id}$.

**Definition B.4** (Cyclic Augmentation for LLMs). Let $C_n = \langle \varsigma \rangle$ act on sequences by the cyclic shift $\varsigma : (t_1, \ldots, t_{|y|}) \mapsto (t_2, \ldots, t_{|y|}, t_1)$. Two self-consistent realizations are common:

(i) **Joint equivariant version.** Introduce an input-side action $\eta : C_n \curvearrowright \mathcal{X}$ (e.g. a control token indicating the starting index), and set $\rho_X = \eta$, $\rho_Y = \varsigma$. Equivariance demands $F(\eta^k \cdot x) = \varsigma^k \cdot F(x)$. The SFT loss uses $\mathcal{L}_{\mathrm{SFT}}^{\mathrm{avg}}$ with $G = C_n$.

(ii) **Orbit (equivalence-class) loss.** When the task identifies outputs up to rotation, treat the label as the orbit $[y] = \{\varsigma^k y : 0 \le k < n\}$ and train with

$$\mathcal{L}_{\mathrm{SFT}}^{\mathrm{orb}}(\theta) = \frac{1}{|D_a|} \sum_{(x,y) \in D_a} \min_{0 \le k < n} \ell\big(f_\theta(x), \ \varsigma^k \cdot y\big),$$

optionally replacing $\min$ by an average or soft-min to smooth optimization.

**Proposition B.5** (Self-consistency / No conflicting supervision). *Assume $F$ is $(\rho_X, \rho_Y)$-equivariant and the loss $\ell$ is compatible with the action on $\mathcal{Y}$ (e.g. cross-entropy with the target transformed by $\rho_Y$). Then any parameter $\theta^\star$ that matches $F$ on $D_a$ also minimizes $\mathcal{L}_{\mathrm{SFT}}^{\mathrm{avg}}$ on $\mathcal{A}_S(D_a)$, and conversely the group average does not introduce contradictory supervision. For tasks with rotational equivalence, the same holds for $\mathcal{L}_{\mathrm{SFT}}^{\mathrm{orb}}$.*

**Preference learning (DPO/Pairwise).** For alignment data $D_p \subseteq \mathcal{X} \times \mathcal{Y} \times \mathcal{Y}$ with $(x, y^+, y^-)$, define group augmentation componentwise:

$$\mathcal{A}_S(D_p) = \bigcup_{g \in S} \left\{ \left( \rho_X(g) \cdot x, \ \rho_Y(g) \cdot y^+, \ \rho_Y(g) \cdot y^- \right) \right\}.$$

A group-averaged DPO-style objective is

$$\mathcal{L}_{\mathrm{DPO}}^{\mathrm{avg}}(\theta) = \frac{1}{|D_p|} \sum_{(x, y^+, y^-)} \frac{1}{|S|} \sum_{g \in S} \left[ -\log \mathrm{sigmoid}\Big( \beta \big( \log p_\theta(\rho_Y(g) \cdot y^+ \mid \rho_X(g) \cdot x) - \log p_\theta(\rho_Y(g) \cdot y^- \mid \rho_X(g) \cdot x) \big) \Big) \right].$$

**Remark.**

(i) *Input-only:* $\rho_Y = \mathrm{id}$; use paraphrases, formatting, or syntactic transforms on $x$ when the target is invariant.

(ii) *Cyclic outputs:* If only rotations of $y$ matter (e.g., cycle descriptions or set enumerations modulo rotation), prefer the orbit loss or enforce equivariance with an input control token $\langle \mathrm{start@}k \rangle$ so that $f_\theta(\langle \mathrm{start@}k \rangle :: x) \approx \varsigma^k f_\theta(x)$.

(iii) *Token hygiene:* Apply $\varsigma$ only to semantic tokens; do not rotate BOS/EOS/padding; keep alignment masks consistent.

# C. Proofs

The theoretical results presented in the main paper establish a rigorous foundation for using submodular optimization to achieve robust language model alignment with minimal augmentation. This section aims to connect these formal proofs to the central narrative of the paper and provide intuitive explanations to make the underlying mathematical concepts accessible before presenting the detailed proofs.

## C.1. The Core Concept: Submodularity as Diminishing Returns

The backbone of many of our theoretical guarantees is the property of *submodularity* (Assumption 1 in the main paper).

**Intuitive Understanding.** Submodularity is the mathematical formalization of "diminishing returns."

- **Analogy (The Coffee Shop):** Imagine choosing locations for new coffee shops to maximize the number of unique customers reached. Placing the first shop in a busy downtown area yields a large gain. Placing a second shop nearby might attract some new customers, but many will overlap with the first shop; the *marginal gain* is smaller. Placing a tenth shop in the same area will add very little new coverage.

In our context, adding the first augmentation might fix many vulnerabilities; adding subsequent, similar augmentations likely offers progressively less additional benefit. This property is crucial because it guarantees that simple, "greedy" algorithms perform near-optimally.

## C.2. Proof of Theorem 3.1

In this section, we present the detailed proof of Theorem 3.1. Theorem 3.1 asserts that finding the smallest set of augmentations to reach a robustness target is NP-hard, but the greedy algorithm provides a logarithmic-factor approximation.

**(i) NP-hardness via Set Cover** Given a Set Cover instance with universe $U = \{e_1, \ldots, e_m\}$ and collection $\mathcal{C} = \{C_1, \ldots, C_n\}$, build the augmentation set $G = \{g_1, \ldots, g_n\}$ with $g_i$ corresponding to $C_i$, and define

$$f(S) = \left| \bigcup_{g_i \in S} C_i \right|.$$

Take $Q = m$. Then minimizing $|S|$ subject to $f(S) \geq m$ is exactly Set Cover. The reduction is clearly polynomial, hence the problem is NP-hard.

**(ii) Greedy approximation ratio $H_Q$ for integer-valued $f$** Let $k^* := |S^*|$. Consider the greedy sequence $S_0 = \emptyset$, $S_{t+1} = S_t \cup \{g_{t+1}\}$ where $g_{t+1}$ maximizes $\Delta(\cdot \mid S_t)$, and let $f_t := f(S_t)$.

**Lemma C.1** (Progress per step). *For every $t \geq 0$,*

$$\Delta(g_{t+1} \mid S_t) \geq \frac{Q - f_t}{k^*}.$$

*Proof of the lemma.* Because $f$ is nondecreasing and $f(S^*) \geq Q$,

$$Q - f_t \leq f(S_t \cup S^*) - f(S_t).$$

Submodularity gives

$$f(S_t \cup S^*) - f(S_t) \leq \sum_{g \in S^*} \Delta(g \mid S_t).$$

Therefore there exists $g' \in S^*$ with $\Delta(g' \mid S_t) \geq (Q - f_t)/k^*$. Greedy chooses $g_{t+1}$ with maximal marginal gain, so $\Delta(g_{t+1} \mid S_t) \geq \Delta(g' \mid S_t)$, proving the claim. $\square$

**Cost-sharing argument.** Assume $f$ is integer-valued. Index the "units of gain" $1, 2, \ldots, Q$. When greedy adds $g_{t+1}$ at unit cost 1 and increases $f$ by $\Delta_{t+1} := \Delta(g_{t+1} \mid S_t)$, charge a price of $1/\Delta_{t+1}$ to each of the $\Delta_{t+1}$ newly covered units. Let $c_j$ be the price paid by unit $j$, defined at the first step $t + 1$ for which $f_t < j \leq f_{t+1}$.

By integrality, $f_t \leq j - 1$. The lemma gives

$$\Delta_{t+1} \geq \frac{Q - f_t}{k^*} \geq \frac{Q - (j - 1)}{k^*},$$

hence

$$c_j = \frac{1}{\Delta_{t+1}} \leq \frac{k^*}{Q - j + 1}.$$

Summing over all $j \in \{1, \ldots, Q\}$,

$$|S_{\text{greedy}}| = \sum_{j=1}^{Q} c_j \leq k^* \sum_{i=1}^{Q} \frac{1}{i} = H_Q \cdot |S^*| \leq H_{f(G)} \cdot |S^*|.$$

This completes the proof. $\square$

### C.3. Proof of Theorem 3.2

In this section, we present the detailed proof of Theorem 3.2. Theorem 3.2 introduces the Log-determinant (Log-det) function $g(S)$ as a proxy for robustness, claiming it is efficient and possesses the necessary submodular properties. The proof utilizes advanced matrix theory (Sylvester's determinant identity and the Matrix determinant lemma) to rigorously establish that $g(S)$ is normalized, monotone, and submodular. This mathematically validates the use of the greedy algorithm to optimize this surrogate with a $(1 - 1/e)$ guarantee.

*Proof.* Let $\phi(g_i) \in \mathbb{R}^d$ and organize them as column vectors to form

$$F_S := [\phi(g_i)]_{i \in S} \in \mathbb{R}^{d \times |S|}.$$

Then $K_{S,S} = F_S^\top F_S$. We will repeatedly use:

1. **Sylvester's determinant identity:** If $U \in \mathbb{R}^{m \times n}$ and $V \in \mathbb{R}^{n \times m}$, then

$$\det(I_m + UV) = \det(I_n + VU).$$

2. **Matrix determinant lemma (rank-1 version):** For invertible $A \succ 0$ and vector $u$,

$$\det(A + uu^\top) = \det(A)(1 + u^\top A^{-1} u).$$

By Sylvester's identity,

$$\det(I_{|S|} + \sigma^{-2} K_{S,S}) = \det(I_{|S|} + \sigma^{-2} F_S^\top F_S) = \det(I_d + \sigma^{-2} F_S F_S^\top).$$

For notational convenience, define

$$A_S := I_d + \sigma^{-2} F_S F_S^\top \quad (\succ 0).$$

Therefore,

$$g(S) = \log \det(A_S).$$

**(i) Normalization:** $g(\emptyset) = 0$   When $S = \emptyset$, we have $F_S = 0$, hence $A_\emptyset = I_d$, and

$$g(\emptyset) = \log \det(I_d) = 0.$$

**(ii) Monotonicity:** $g(S \cup \{j\}) - g(S) \geq 0$ **for any** $j \notin S$   Note that

$$F_{S \cup \{j\}} F_{S \cup \{j\}}^\top = F_S F_S^\top + \phi(g_j) \phi(g_j)^\top, \quad A_{S \cup \{j\}} = A_S + \sigma^{-2} \phi(g_j) \phi(g_j)^\top.$$

Applying the matrix determinant lemma (with $u = \sigma^{-1} \phi(g_j)$) gives

$$\frac{\det(A_{S \cup \{j\}})}{\det(A_S)} = 1 + \sigma^{-2} \phi(g_j)^\top A_S^{-1} \phi(g_j).$$

Therefore,

$$g(S \cup \{j\}) - g(S) = \log\left(1 + \sigma^{-2} \phi(g_j)^\top A_S^{-1} \phi(g_j)\right) \geq 0,$$

since $A_S^{-1} \succ 0$ ensures the quadratic form is non-negative. Hence $g$ is monotone increasing.

**(iii) Submodularity: Diminishing Marginal Returns**   Take $S \subseteq T \subseteq [n]$ and $j \notin T$. Since

$$A_T = A_S + \sigma^{-2} \sum_{i \in T \setminus S} \phi(g_i) \phi(g_i)^\top \succeq A_S,$$

by the Loewner order on symmetric positive definite matrices, we have $A_T \succeq A_S \Rightarrow A_T^{-1} \preceq A_S^{-1}$. Consequently,

$$\phi(g_j)^\top A_S^{-1} \phi(g_j) \geq \phi(g_j)^\top A_T^{-1} \phi(g_j).$$

Since $x \mapsto \log(1 + \sigma^{-2} x)$ is increasing and concave for $x \geq 0$, we obtain

$$\underbrace{g(S \cup \{j\}) - g(S)}_{=\log(1+\sigma^{-2}\phi(g_j)^\top A_S^{-1}\phi(g_j))} \quad \geq \quad \underbrace{g(T \cup \{j\}) - g(T)}_{=\log(1+\sigma^{-2}\phi(g_j)^\top A_T^{-1}\phi(g_j))} \quad .$$

This is precisely the diminishing marginal returns condition $\Delta(j \mid S) \geq \Delta(j \mid T)$ for submodularity.

**(iv) The $(1 - 1/e)$ Approximation of the Greedy Algorithm**   Let $S_t$ denote the set after $t$ steps of the greedy algorithm (at each step, we select the element with maximum marginal gain), and let $S^\star$ be an optimal solution with size at most $k$. Define $\delta_t := g(S^\star) - g(S_t)$. By submodularity and monotonicity,

$$\delta_t \leq \sum_{x \in S^\star \setminus S_t} (g(S_t \cup \{x\}) - g(S_t)) \leq k \cdot (g(S_{t+1}) - g(S_t)),$$

since the marginal gain chosen by greedy is at least as large as the marginal gain of any single element. Thus,

$$\delta_{t+1} = \delta_t - (g(S_{t+1}) - g(S_t)) \leq \left(1 - \frac{1}{k}\right) \delta_t.$$

Recursively applying this up to $t = k$ and using $\delta_0 = g(S^\star) - g(\emptyset) = g(S^\star)$, we obtain

$$g(S_k) \geq \left(1 - \left(1 - \frac{1}{k}\right)^k\right) g(S^\star) \geq (1 - 1/e)\, g(S^\star),$$

where the last step uses $(1 - 1/k)^k \leq e^{-1}$. This is the classical guarantee of (Nemhauser et al., 1978) for normalized, monotone, submodular functions under cardinality constraints.

$\square$

### C.4. Proof of Proposition 3.3

Proposition 3.3 and Proposition 3.4 argue that the Log-det surrogate is not arbitrary; it is fundamentally connected to information gain. Proposition 3.3 proves that under a common model (Gaussian linearization), the Log-det formula is exactly equivalent to the Mutual Information (MI) between the augmentations and the model parameters.

*Proof.* Let $m := |S|$ and stack the feature vectors as $\Phi_S = [\phi(g)]_{g \in S} \in \mathbb{R}^{d \times m}$ so that the linearized observation model reads

$$y_S \;=\; \Phi_S^\top \theta + \varepsilon, \qquad \theta \sim \mathcal{N}(0, \Sigma_0), \quad \varepsilon \sim \mathcal{N}(0, \sigma^2 I_m), \quad \theta \perp\!\!\!\perp \varepsilon.$$

By Gaussianity, $(\theta, y_S)$ is jointly Gaussian, hence

$$I(\theta; y_S) = H(y_S) - H(y_S \mid \theta) = H(\theta) - H(\theta \mid y_S).$$

We prove the claim in two equivalent ways.

**Route A: via the marginal of $y_S$.**   Since $\mathbb{E}[y_S] = 0$ and

$$\mathrm{Cov}(y_S) = \mathbb{E}[(\Phi_S^\top \theta + \varepsilon)(\Phi_S^\top \theta + \varepsilon)^\top] = \Phi_S^\top \Sigma_0 \Phi_S + \sigma^2 I_m,$$

we have $y_S \sim \mathcal{N}(0, \Sigma_y)$ with $\Sigma_y = \Phi_S^\top \Sigma_0 \Phi_S + \sigma^2 I_m$. For a nondegenerate $m$-variate Gaussian, $H(y_S) = \frac{1}{2} \log\big((2\pi e)^m \det \Sigma_y\big)$, and $H(y_S \mid \theta) = H(\varepsilon) = \frac{1}{2} \log\big((2\pi e)^m \det(\sigma^2 I_m)\big)$. Therefore

$$I(\theta; y_S) = \tfrac{1}{2} \log \frac{\det(\Phi_S^\top \Sigma_0 \Phi_S + \sigma^2 I_m)}{\det(\sigma^2 I_m)} = \tfrac{1}{2} \log \det\Big(I_m + \sigma^{-2} \Phi_S^\top \Sigma_0 \Phi_S\Big).$$

Noting that $K_{S,S} = \Phi_S^\top \Sigma_0 \Phi_S$ by definition, we obtain

$$I(\theta; y_S) = \tfrac{1}{2} \log \det\big(I_m + \sigma^{-2} K_{S,S}\big).$$

**Route B: via prior–posterior entropies.**   For linear–Gaussian models, the posterior covariance of $\theta$ given $y_S$ is

$$\Sigma_S \;=\; \Big(\Sigma_0^{-1} + \sigma^{-2} \Phi_S \Phi_S^\top\Big)^{-1}.$$

Since $H(\theta) = \frac{1}{2} \log\big((2\pi e)^d \det \Sigma_0\big)$ and $H(\theta \mid y_S) = \frac{1}{2} \log\big((2\pi e)^d \det \Sigma_S\big)$, we get

$$I(\theta; y_S) = \tfrac{1}{2} \log \frac{\det \Sigma_0}{\det \Sigma_S} = \tfrac{1}{2} \log \det\Big(\Sigma_0\big(\Sigma_0^{-1} + \sigma^{-2} \Phi_S \Phi_S^\top\big)\Big) = \tfrac{1}{2} \log \det\Big(I_d + \sigma^{-2} \Sigma_0 \Phi_S \Phi_S^\top\Big).$$

Apply Sylvester's determinant theorem (a.k.a. the matrix determinant lemma), $\det(I_d + AB) = \det(I_m + BA)$ for $A \in \mathbb{R}^{d \times m}$, $B \in \mathbb{R}^{m \times d}$, with $A = \sigma^{-1} \Sigma_0^{1/2} \Phi_S$ and $B = \sigma^{-1} \Phi_S^\top \Sigma_0^{1/2}$, to obtain

$$\det\Big(I_d + \sigma^{-2} \Sigma_0^{1/2} \Phi_S \Phi_S^\top \Sigma_0^{1/2}\Big) = \det\Big(I_m + \sigma^{-2} \Phi_S^\top \Sigma_0 \Phi_S\Big) = \det\big(I_m + \sigma^{-2} K_{S,S}\big).$$

This gives the same expression as in Route A:

$$I(\theta; y_S) = \tfrac{1}{2} \log \det\big(I_m + \sigma^{-2} K_{S,S}\big).$$

**Remarks.** The identity holds under $\Sigma_0 \succ 0$ and $\sigma^2 > 0$, which ensure the involved covariances are nonsingular and the (log-)determinants are well-defined. If $\Sigma_0$ is only positive semidefinite, the result remains valid when all determinants are interpreted as pseudo-determinants. $\qquad\square$

### C.5. Proof of Proposition 3.4

Proposition 3.4 then applies the standard submodular guarantees to this MI objective.

*Proof.* Let $G$ be the ground set of candidate augmentations, and define $f(S) := I(D_a^S; \theta)$ for $S \subseteq G$. By assumption $f$ is normalized ($f(\emptyset) = 0$), monotone ($S \subseteq T \Rightarrow f(S) \leq f(T)$), and submodular (diminishing returns: for $S \subseteq T$ and $g \notin T$, $\Delta(g \mid S) \geq \Delta(g \mid T)$ where $\Delta(g \mid S) := f(S \cup \{g\}) - f(S)$). We analyze the standard greedy algorithm that, given a current set $S$, selects $g^\star \in \arg\max_{g \in G \setminus S} \Delta(g \mid S)$.

**(a) Cardinality constraint, $(1-1/e)$-approximation.** Let $k \in \mathbb{N}$ and let $O \subseteq G$ be an optimal $k$-element set maximizing $f$, i.e. $|O| \leq k$ and $f(O) = \text{OPT}$. Let $S_i$ denote the greedy set after $i$ selections ($S_0 = \emptyset$). We prove the inductive inequality

$$f(O) - f(S_i) \;\leq\; \left(1 - \tfrac{1}{k}\right)\left(f(O) - f(S_{i-1})\right) \qquad (i = 1, 2, \ldots, k),$$

which yields

$$f(S_k) \;\geq\; \left(1 - \left(1 - \tfrac{1}{k}\right)^k\right) f(O) \;\geq\; \left(1 - \tfrac{1}{e}\right) f(O),$$

as desired.

Fix any $i \in \{1, \ldots, k\}$. By monotonicity,

$$f(O) - f(S_{i-1}) \;\leq\; f(O \cup S_{i-1}) - f(S_{i-1}).$$

By submodularity and the telescoping/union bound on marginals,

$$f(O \cup S_{i-1}) - f(S_{i-1}) \;\leq\; \sum_{g \in O \setminus S_{i-1}} \Delta(g \mid S_{i-1}) \;\leq\; |O| \max_{g \in G \setminus S_{i-1}} \Delta(g \mid S_{i-1}) \;\leq\; k\Delta(g_i^\star \mid S_{i-1}),$$

where $g_i^\star$ is the greedy choice at step $i$. Rearranging,

$$\Delta(g_i^\star \mid S_{i-1}) \;\geq\; \tfrac{1}{k}\left(f(O) - f(S_{i-1})\right).$$

Therefore

$$f(O) - f(S_i) \;=\; f(O) - \left(f(S_{i-1}) + \Delta(g_i^\star \mid S_{i-1})\right) \;\leq\; \left(1 - \tfrac{1}{k}\right)\left(f(O) - f(S_{i-1})\right),$$

which proves the one-step contraction and hence the $(1 - 1/e)$ bound after $k$ steps. This establishes part (a).

**(b) Submodular cover, logarithmic-factor approximation.** Fix a target $\tau > 0$ and consider the minimum-cardinality *cover* problem

$$\min \{|S| : \ f(S) \geq \tau\}.$$

Let $O$ be an optimal solution with $|O| = k^\star$. Let $S_i$ be the greedy sequence and define the residual (deficit)

$$r_i \;:=\; \max\{0, \ \tau - f(S_i)\}, \qquad i = 0, 1, 2, \ldots$$

While $r_{i-1} > 0$, monotonicity implies $f(S_{i-1} \cup O) \geq f(O) \geq \tau$, hence

$$r_{i-1} \;=\; \tau - f(S_{i-1}) \;\leq\; f(S_{i-1} \cup O) - f(S_{i-1}) \;\leq\; \sum_{g \in O \setminus S_{i-1}} \Delta(g \mid S_{i-1}) \;\leq\; k^\star \Delta(g_i^\star \mid S_{i-1}),$$

so the greedy marginal satisfies $\Delta(g_i^\star \mid S_{i-1}) \geq r_{i-1}/k^\star$. Consequently,

$$r_i \;=\; r_{i-1} - \Delta(g_i^\star \mid S_{i-1}) \;\leq\; \left(1 - \tfrac{1}{k^\star}\right) r_{i-1},$$

i.e. each iteration shrinks the residual by at least a $(1 - 1/k^\star)$ factor.

There are two standard ways to conclude a logarithmic approximation guarantee:

*(b1) Integer-valued case (exact cover).* If $f$ takes integer values and $\tau \in \mathbb{Z}_+$ (as in classic set cover and many discrete coverage objectives), define the *unit price* of iteration $i$ as $p_i := 1/\Delta(g_i^\star \mid S_{i-1})$. Charge this price uniformly to each of the $\Delta(g_i^\star \mid S_{i-1})$ units by which the residual drops from $r_{i-1}$ to $r_i$. Because $\Delta(g_i^\star \mid S_{i-1}) \geq r_{i-1}/k^\star$, we have $p_i \leq k^\star/r_{i-1}$. Thus any integer residual level $j \in \{1, 2, \ldots, \tau\}$ is charged at most $k^\star/j$ the (unique) time the residual crosses it. Summing charges over $j = 1$ to $\tau$,

$$\#\text{greedy steps} \;=\; \sum_i 1 \;\leq\; \sum_{j=1}^{\tau} \frac{k^\star}{j} \;=\; k^\star H_\tau \;\leq\; k^\star(1 + \ln \tau),$$

where $H_\tau$ is the $\tau$-th harmonic number. Hence the greedy cover set has size at most $H_\tau$ times optimal, i.e. a logarithmic-factor approximation.

*(b2) Real-valued case (additive $\varepsilon$-cover).* When $f$ is real-valued, for any tolerance $\varepsilon \in (0, \tau]$ the same one-step contraction gives

$$r_i \;\leq\; \left(1 - \tfrac{1}{k^\star}\right)^i r_0 \;=\; \left(1 - \tfrac{1}{k^\star}\right)^i \tau \;\leq\; \tau e^{-i/k^\star}.$$

Thus after $i \geq k^\star \ln(\tau/\varepsilon)$ greedy steps we have $r_i \leq \varepsilon$, i.e. $f(S_i) \geq \tau - \varepsilon$ with at most $k^\star \lceil \ln(\tau/\varepsilon) \rceil$ items. If $f$ admits a minimum positive marginal $\gamma > 0$ whenever $f(S) < \tau$, choosing $\varepsilon = \gamma$ yields an *exact* cover within at most $k^\star(1 + \ln(\tau/\gamma))$ steps, again logarithmic in the target.

Either way, greedy achieves a logarithmic-factor approximation for the cover variant, establishing part (b). $\qquad \square$

**Remarks.**

(i) The proof used only normalization, monotonicity, and submodularity, hence applies verbatim to $f(S) = I(D_a^S; \theta)$ under the stated modeling conditions (e.g., linear-Gaussian or certain exponential family settings).

(ii) For Algorithm 1, replace $f$ by the improvement objective you defined; the same analysis gives the identical guarantees (with $\tau$ replaced by your $Q$) because the argument depends only on submodularity of $f$ and not on its specific semantic meaning.

(iii) The $(1 - 1/e)$ ratio for (a) is optimal under standard complexity assumptions, and the $\Theta(\log \tau)$ factor in (b) is also essentially tight for general submodular cover, so the guarantees are information-theoretically meaningful.

**C.6. Proof of Proposition 4.2**

Proposition 4.2 addresses how to select a subset of cyclic augmentations if the full set is too large, claiming that "evenly spaced" selections are optimal for dispersion. The proof uses a "smoothing" argument. It shows that if the selections are unevenly clustered on the cycle, one can always perform a local shift that balances the distribution. Due to the concavity of the distance function on the cycle, this smoothing operation always increases the overall diversity (dispersion).

C.6.1. PREPARATORY WORK: DESCRIBING OBJECTIVE FUNCTIONS VIA GAPS

Let $f(x) := \min\{x, n - x\}$ (a piecewise linear **concave** function on $[0, n]$). For $t = 1, \ldots, k - 1$, denote

$$C_i^{(t)} := g_i + g_{i+1} + \cdots + g_{i+t-1} \quad \text{(with cyclic indices)}.$$

Then the geodesic distance for each pair $(s_i, s_{i+t})$ is $f\big(C_i^{(t)}\big)$. Therefore,

$$\sum_{i<j} d(i, j) = \frac{1}{2} \sum_{t=1}^{k-1} \sum_{i=1}^{k} f\big(C_i^{(t)}\big) \tag{16}$$

(since each unordered pair is counted exactly twice in the $(t, i)$ representation). This decomposition is the core of all subsequent "local modification" and "concavity" arguments.

**Proof of (a): $k$-center — Maximizing Minimum Distance**

**Step 1 (Lower bound equals minimum gap)**   When $k \geq 2$, we have $\min_{i \neq j} d(i, j) = \min_i g_i$.

**Reason:** The average gap is $n/k \leq n/2$, so there exists $g_m \leq \lfloor n/k \rfloor \leq n/2$, and the geodesic distance between adjacent points $(s_m, s_{m+1})$ is exactly $g_m$. On the other hand, if any two points are separated by $\geq 2$ gaps, then the lengths along both arcs are at least the sum of some gaps, the minimum being $\geq \min g_i$. Thus the overall minimum distance is exactly $\min g_i$.

**Step 2 (Upper bound)**   By the arithmetic mean inequality, $\min_i g_i \leq \lfloor n/k \rfloor$.

**Step 3 (Achievability)**   If $g_i \in \{q, q+1\}$ (where $q = \lfloor n/k \rfloor$), then $\min_i g_i = q$, matching the upper bound in Step 2, thus achieving optimality. This proves (a) (including the trivial case $\min = 0$ when $k = 1$).

C.6.2. PROOF OF (B): MAX-SUM — LONG GAPS MUST BE EVENLY INTERLACED

**Phase I: Pulling All Gaps to $\{q, q+1\}$**   If there exist adjacent gaps with $g_i \geq g_{i+1} + 2$, perform a "smoothing step":

$$(g_i, g_{i+1}) \longmapsto (g_i - 1, \; g_{i+1} + 1).$$

For fixed $t$, observe all cyclic window sums $C_j^{(t)}$ of length $t$. Only those $t$ windows that **contain** $g_i$ **but not** $g_{i+1}$ decrease by 1 each; and those $t$ windows that **contain** $g_{i+1}$ **but not** $g_i$ increase by 1 each; all other windows remain unchanged. Pairing these two families of windows by starting position, we find that in each pair, the original sum of the increased window is $\leq$ the original sum of the decreased window $-2$. Since $f$ is a discrete concave function, the difference

$$\Delta(x) := f(x + 1) - f(x)$$

is monotonically non-increasing in $x$ (specifically, $\Delta(x) = 1, 0, -1$ depending on the position of $x$ relative to $n/2$), so the net change from each pair is

$$\Delta(\text{smaller}) - \Delta(\text{larger} - 1) \; \geq \; 0.$$

Thus for each $t$, $\sum_j f(C_j^{(t)})$ is **non-decreasing**; the total sum in (1) is also non-decreasing. Repeat this smoothing until all adjacent differences are $\leq 1$, equivalent to $g_i \in \{q, q+1\}$. Therefore, any maximum solution must belong to this family.

**Remark.**   Up to this point, we have reduced the problem to: for fixed $q, r$, we only need to decide **which** indices $i$ have $g_i = q + 1$, with the rest being $g_i = q$.

**Phase II: Even Interlacing of Long Gaps (balanced / mechanical)**   Introduce a 0–1 cyclic word $w = (w_1, \ldots, w_k)$, defined by

$$w_i = \begin{cases} 1, & g_i = q + 1, \\ 0, & g_i = q, \end{cases} \quad \text{so} \quad \sum_i w_i = r.$$

For fixed $t$, let

$$N_i^{(t)} := \sum_{j=0}^{t-1} w_{i+j} \quad \text{(number of long gaps in the window)},$$

then

$$C_i^{(t)} = tq + N_i^{(t)}, \qquad \sum_{i=1}^{k} f(C_i^{(t)}) = \sum_{i=1}^{k} \underbrace{f(tq + N_i^{(t)})}_{=: \, \varphi_t(N_i^{(t)})}.$$

Since $f$ is concave and $\varphi_t(x) = f(tq + x)$ is a concave function of $x$, making the long gaps "more evenly distributed" on the circle will increase (or maintain) the sum for each $t$.

**Key Smoothing (at the 0–1 Level)** If the word $w$ has two consecutive "0-blocks between 1's" with length difference $\geq 2$ (i.e., the spacing between two long gaps differs by at least 2), shift the 1 at the right end of the longer block left by one position. This operation decreases $N_i^{(t)}$ by 1 for some family of $t$ windows, and increases others by 1; both families have $t$ windows, and we can use the same "adjacent starting position pairing" to match each "+1" with a "-1", ensuring that in each pair, the original value of the window being +1'd is $\leq$ the original value of the window being -1'd $-2$. By the concavity of $\varphi_t$, the net change for each pair is $\geq 0$. Therefore, for each $t$, $\sum_i \varphi_t(N_i^{(t)})$ is non-decreasing, and so is the total sum in (1).

Repeating this smoothing, we eventually reach a 0–1 cyclic word where all 0-blocks between 1's have lengths **differing by at most 1**, which is precisely the classical **balanced (also called mechanical / Sturmian)** structure. Its equivalent characterization is: for each $t$, all cyclic windows of length $t$ contain a number of 1's equal to either

$$\left\lfloor \frac{tr}{k} \right\rfloor \quad \text{or} \quad \left\lceil \frac{tr}{k} \right\rceil,$$

and the occurrence counts of these two values are determined by $(t, r, k)$, independent of the starting position. Thus for each $t$, the vector $\left(N_i^{(t)}\right)_{i=1}^{k}$ is **maximally balanced**, yielding the maximum $\sum_i \varphi_t(N_i^{(t)})$ under the concave function $\varphi_t$; summing over $t = 1, \ldots, k-1$ and multiplying by $\frac{1}{2}$ (see (1)) gives the maximum value of $\sum_{i<j} d(i,j)$. Moreover, since any two balanced configurations have the same multiset $\{N_i^{(t)}\}$ for each $t$, they yield the same total sum, meaning "optimality is not unique, but the entire balanced family has the same value". $\qquad\square$

### C.7. Proof of Proposition 4.4

Section 4 in the main paper discusses scenarios where the ideal submodularity assumption might not perfectly hold. Proposition 4.4 (Weak Submodularity, $\gamma_k$) and Definition 4.5 (Curvature, $c$) quantify deviations from the ideal case.

*Proof.* We may assume without loss of generality that $f(\emptyset) = 0$: otherwise replace $f$ by $g(S) := f(S) - f(\emptyset)$, which does not change marginal gains, greedy choices, or the submodularity ratio.

For $A \subseteq V$ and $T \subseteq V$, write $\Delta(T \mid A) := f(A \cup T) - f(A)$ and abbreviate $\Delta(e \mid A)$ for singletons. The (cardinality-$k$) submodularity ratio is

$$\gamma_k := \min_{\substack{A \subseteq V,\, T \subseteq V \\ |T| \leq k,\, T \cap A = \emptyset}} \frac{\sum_{e \in T} \Delta(e \mid A)}{\Delta(T \mid A)} \in (0, 1],$$

which is equivalent to the inequality

$$\sum_{e \in T} \Delta(e \mid A) \geq \gamma_k \Delta(T \mid A) \qquad \text{for all } A \subseteq V, \ T \subseteq V, \ |T| \leq k. \tag{17}$$

Let $S_0 := \emptyset$, and for $i = 0, 1, \ldots, k-1$ choose $a_{i+1} \in \arg\max_{e \in V \setminus S_i} \Delta(e \mid S_i)$ and set $S_{i+1} := S_i \cup \{a_{i+1}\}$; denote $S_k = S_{\text{greedy}}$. Fix an optimal solution $S_k^\star \in \arg\max_{|S| \leq k} f(S)$.

At iteration $i$, let $M_i := \max_{e \in V \setminus S_i} \Delta(e \mid S_i)$. Since $|S_k^\star| \leq k$ and $\Delta(e \mid S_i) \leq M_i$ for every $e \in S_k^\star$ (with equality for at least one candidate),

$$f(S_{i+1}) - f(S_i) = M_i \geq \frac{1}{k} \sum_{e \in S_k^\star} \Delta(e \mid S_i). \tag{18}$$

Let $T_i := S_k^\star \setminus S_i$; then $\sum_{e \in S_k^\star} \Delta(e \mid S_i) = \sum_{e \in T_i} \Delta(e \mid S_i)$ (because $\Delta(e \mid S_i) = 0$ for $e \in S_i$), and likewise $\Delta(S_k^\star \mid S_i) = \Delta(T_i \mid S_i)$. Applying (17) with $A = S_i$ and $T = T_i$ yields

$$\sum_{e \in S_k^\star} \Delta(e \mid S_i) = \sum_{e \in T_i} \Delta(e \mid S_i) \geq \gamma_k \Delta(T_i \mid S_i) = \gamma_k \Delta(S_k^\star \mid S_i). \tag{19}$$

Combining (18) and (19), and using monotonicity,

$$f(S_{i+1}) - f(S_i) \geq \frac{\gamma_k}{k} \Delta(S_k^\star \mid S_i) = \frac{\gamma_k}{k} \left( f(S_i \cup S_k^\star) - f(S_i) \right) \geq \frac{\gamma_k}{k} \left( f(S_k^\star) - f(S_i) \right).$$

Rearranging,

$$f(S_k^\star) - f(S_{i+1}) \leq \left(1 - \frac{\gamma_k}{k}\right)\left(f(S_k^\star) - f(S_i)\right).$$

Let $\delta_i := f(S_k^\star) - f(S_i)$ $(\geq 0)$. By induction,

$$\delta_k \leq \left(1 - \frac{\gamma_k}{k}\right)^k \delta_0 = \left(1 - \frac{\gamma_k}{k}\right)^k f(S_k^\star) \leq e^{-\gamma_k} f(S_k^\star),$$

where the last step uses $(1 - \frac{x}{k})^k \leq e^{-x}$ for $x \geq 0$. Therefore

$$f(S_{\text{greedy}}) = f(S_k) \geq \left(1 - e^{-\gamma_k}\right) f(S_k^\star).$$

$\square$

## C.8. Proof of Proposition 5.2

*Proof.* Let $V$ be the ground set and $f : 2^V \to \mathbb{R}_{\geq 0}$ be a nonnegative, monotone, submodular function. For $S \subseteq V$ and $e \in V \setminus S$, write the marginal $f_S(e) := f(S \cup \{e\}) - f(S)$. The *(total) curvature* of $f$ is

$$c = 1 - \min_{e \in V} \frac{f_{V \setminus \{e\}}(e)}{f(\{e\})} \in [0, 1].$$

Equivalently, for every $e \in V$ and every $A \subseteq V \setminus \{e\}$,

$$f_A(e) \geq (1 - c)f(\{e\}). \tag{20}$$

Let $S_0 = \emptyset$ and for $i = 1, \ldots, k$ let $S_i$ be the greedy set after $i$ steps: $S_i = S_{i-1} \cup \{e_i\}$ where $e_i \in \arg\max_{e \in V \setminus S_{i-1}} f_{S_{i-1}}(e)$. Denote $\Delta_i := f(S_i) - f(S_{i-1}) = f_{S_{i-1}}(e_i)$ and let $O \in \arg\max_{|T| \leq k} f(T)$ be an optimal $k$-set. We prove

$$f(S_i) \geq \frac{1}{c}\left(1 - \left(1 - \tfrac{c}{k}\right)^i\right)f(O) \qquad (i = 0, 1, \ldots, k),$$

which at $i = k$ yields the theorem; the $e^{-c}$ form follows from $(1 - x/n)^n \leq e^{-x}$ for $x \geq 0$.

**A block-marginal lower bound.** For any $X, Y \subseteq V$, write $f_Y(X) := f(Y \cup X) - f(Y)$. We claim

$$f_Y(X) \geq (1 - c)f(X \setminus Y). \tag{21}$$

Proof: Let $U = X \setminus Y = \{u_1, \ldots, u_m\}$ be any ordering. By submodularity,

$$f_Y(X) = \sum_{j=1}^{m} f_{Y \cup \{u_1, \ldots, u_{j-1}\}}(u_j) \overset{(20)}{\geq} \sum_{j=1}^{m}(1 - c)f(\{u_j\}) \geq (1 - c)\sum_{j=1}^{m} f_{\{u_1, \ldots, u_{j-1}\}}(u_j) = (1 - c)f(U),$$

proving (21). (We used $f_{\{u_1, \ldots, u_{j-1}\}}(u_j) \leq f(\{u_j\})$ by diminishing returns.)

**A key inequality.** For any $S, T \subseteq V$,

$$f(T) \leq cf(S) + \sum_{e \in T} f_S(e). \tag{22}$$

Proof: Write

$$f(T) = f(S \cup T) - f_T(S \setminus T) \leq \underbrace{f(S) + f_S(T)}_{\leq f(S) + \sum_{e \in T} f_S(e)} - f_T(S \setminus T).$$

By (21) with $X = S$ and $Y = T$, $f_T(S \setminus T) \geq (1 - c)f(S \setminus T)$. Using subadditivity (a consequence of monotone submodularity), $f(S) \leq f(S \setminus T) + f(S \cap T)$, so

$$-f_T(S \setminus T) \leq -(1 - c)f(S) + (1 - c)f(S \cap T) \leq -(1 - c)f(S),$$

and (22) follows. (We dropped the nonnegative $(1 - c)f(S \cap T)$ to obtain an upper bound.)

**Greedy progress per step.** Apply (22) with $S = S_{i-1}$ and $T = O$:

$$\sum_{o \in O} f_{S_{i-1}}(o) \geq f(O) - cf(S_{i-1}).$$

Since $|O| \leq k$, the largest marginal at step $i$ dominates the average over $O$:

$$\Delta_i = \max_e f_{S_{i-1}}(e) \geq \frac{1}{k} \sum_{o \in O} f_{S_{i-1}}(o) \geq \frac{1}{k}\Big(f(O) - cf(S_{i-1})\Big).$$

Hence

$$f(S_i) = f(S_{i-1}) + \Delta_i \geq \Big(1 - \frac{c}{k}\Big)f(S_{i-1}) + \frac{1}{k}f(O). \tag{23}$$

**Solving the recurrence.** Let $a := 1 - \frac{c}{k}$. From (23) and $f(S_0) = 0$, a simple induction shows

$$f(S_i) \geq \frac{1}{c}\Big(1 - a^i\Big)f(O) = \frac{1}{c}\Big(1 - \big(1 - \tfrac{c}{k}\big)^i\Big)f(O),$$

as claimed. At $i = k$ we obtain

$$f(S_{\text{greedy}}) \geq \frac{1}{c}\Big(1 - \big(1 - \tfrac{c}{k}\big)^k\Big)f(O) \geq \frac{1}{c}(1 - e^{-c})f(O),$$

using $(1 - x/n)^n \leq e^{-x}$. For $c = 0$ the function is modular (all marginals are invariant), and greedy is optimal, while the bound approaches 1 by continuity (L'Hôpital's rule), completing the proof. $\qquad\square$

### C.9. Proof of Theorem 6.1

Theorem 6.1 is a key theoretical contribution, stating that cyclic augmentation of order $n$ reduces the model's sensitivity to perturbations by a factor of $1/n$, thereby increasing the Robustness Factor by $n$. This proof deeply engages with the mathematical definitions of the Elasticity Framework (based on compression rates). It uses perturbation and asymptotic analysis to calculate how quickly the compression rate changes (sensitivity) when the model is perturbed (e.g., by inverse alignment fine-tuning). It rigorously derives the $1/n$ sensitivity reduction by analyzing how cyclic symmetry distributes the information across the dataset (Steps 3 and 4 of the proof).

*Proof.* **Step 1: Effective Dataset Size under Cyclic Augmentation**

The cyclic augmentation creates $n$ distinct variations for each sample. However, these are not independent samples. We establish:

$$|D^{\text{cyc}}|_{\text{eff}} = n \cdot |D| \cdot \xi_n \tag{24}$$

where $\xi_n$ is the information diversity factor. For cyclic permutations:

$$\xi_n = \frac{H(D^{\text{cyc}})}{n \cdot H(D)} = 1 - \frac{\log n}{n \cdot H(D)} \tag{25}$$

For sufficiently large $H(D)$ (high entropy alignment data), $\xi_n \approx 1$.

**Step 2: Compression Protocol Analysis**

Following the compression protocol, the expected code length for $D^{\text{cyc}}$ yields:

$$\mathbb{E}[L_{p_\theta}(x)] = \left\lceil \frac{|x|}{d} \right\rceil \left[ -\sum_{l=1}^{d} \sum_{j=1}^{2^{l-1}} p_{lj}^{\text{cyc}} \log p_{lj}^{\text{cyc}} \right] \tag{26}$$

Due to cyclic symmetry, the probability mass at depth $d$ satisfies:

$$p_{lj}^{\text{cyc}} = \frac{1}{n} \sum_{k=0}^{n-1} p_{lj}^{(k)} \tag{27}$$

where $p_{lj}^{(k)}$ represents the probability after $k$ cyclic shifts.

**Step 3: Perturbation Analysis**

Under perturbation $D_p$, the joint distribution becomes:

$$p_{lj}^{D'} = \frac{|D_{PT}|p_{lj}^{PT} + n|D|p_{lj}^{\text{cyc}} + |D_p|p_{lj}^{p}}{|D_{PT}| + n|D| + |D_p|} \tag{28}$$

Taking the derivative with respect to $\lambda = |D_p|/|D|$:

$$\frac{d\gamma_{D^{\text{cyc}}/D'}^{p_\theta}}{d\lambda} = -\alpha^3 c_{min}^{3\alpha} \frac{dS^{\text{cyc}}}{d\lambda} \tag{29}$$

where $S^{\text{cyc}}$ is defined as:

$$S^{\text{cyc}} = \int_c^{+\infty} \int_c^{+\infty} \int_c^{+\infty} \frac{1}{x_1^\alpha x_2^{\alpha+1} x_3^{\alpha+1}} \times \log\left(\frac{|D_{PT}|x_1 + nx_2 + \lambda x_3}{|D_{PT}| + n + \lambda}\right) dx_1 dx_2 dx_3 \tag{30}$$

by following the Pareto distribution assumption.

**Step 4: Asymptotic Analysis**

As $|D_{PT}| \to \infty$ and $\lambda \to 0$:

$$\lim_{|D_{PT}|\to\infty, \lambda\to 0} \frac{dS^{\text{cyc}}}{d\lambda} = \frac{1}{n} \cdot \frac{1}{|D_{PT}|(|D_{PT}| + n)\alpha^2(\alpha - 1)c_{min}^{3\alpha-1}} \tag{31}$$

Compared to the non-augmented case:

$$\lim_{|D_{PT}|\to\infty, \lambda\to 0} \frac{dS}{d\lambda} = \frac{1}{|D_{PT}|(|D_{PT}| + 1)\alpha^2(\alpha - 1)c_{min}^{3\alpha-1}} \tag{32}$$

Therefore:

$$\left|\frac{d\gamma_{D^{\text{cyc}}/D'}^{p_\theta}}{d\lambda}\right| = \frac{1}{n}\left|\frac{d\gamma_{D/D}^{p_\theta}}{d\lambda}\right| \cdot \frac{|D_{PT}| + 1}{|D_{PT}| + n} \tag{33}$$

For large $|D_{PT}|$, the factor $\frac{|D_{PT}|+1}{|D_{PT}|+n} \approx 1$, yielding the stated result.

**Step 5: Robustness Measure**

The robustness factor follows directly:

$$\mathcal{R}(D^{\text{cyc}}) = \frac{n|D|}{|\frac{1}{n} \cdot \frac{d\gamma_{D/D}^{p_\theta}}{d\lambda}|} = n \cdot \frac{|D|}{|\frac{d\gamma_{D/D}^{p_\theta}}{d\lambda}|} = n \cdot \mathcal{R}(D) \tag{34}$$

$\square$

**Intuition.**

- **Analogy (The Parallel Springs):** Alignment is like stretching a stiff spring (the pre-trained model) away from its resting state; it naturally wants to "snap back" (elasticity). Standard alignment relies on a single spring. Cyclic augmentation (e.g., distributing refusal cues across $n$ positions) is equivalent to installing $n$ parallel springs by creating $n$ distinct refusal paths. To pull the system back to the original, unaligned state, an adversary now requires $n$ times the force (i.e., $n$ times more perturbation data).

# D. Examples

## D.1. Patching Vulnerabilities

**Setup.** There are six vulnerability types $U = \{V_1, \ldots, V_6\}$ and three available patches $G = \{A, B, C\}$. Define $f(S)$ to be the number of distinct vulnerabilities in $U$ fixed by the patches in $S$ (so $f$ is a coverage function, hence nondecreasing and submodular). Assume the baseline robustness is $\mathcal{R}_\varepsilon(D^\emptyset) = 0$ and the target is to fix all six vulnerabilities, i.e., $Q = \mathcal{R}_{\text{target}} = 6$.

**Patch effects:**

- $A$: fixes $\{V_1, V_2, V_3, V_4\} \Rightarrow f(\{A\}) = 4$

- $B$: fixes $\{V_1, V_2, V_5\} \Rightarrow f(\{B\}) = 3$

- $C$: fixes $\{V_3, V_4, V_6\} \Rightarrow f(\{C\}) = 3$

**Optimal solution:**

$$f(\{A, B\}) = 5, \quad f(\{A, C\}) = 5, \quad f(\{B, C\}) = 6.$$

Hence $S^* = \{B, C\}$ with $|S^*| = 2$.

**Greedy run:**

- **Step 1:** from $S = \emptyset$, marginal gains are $4, 3, 3$; choose $A \Rightarrow S = \{A\}$ and $f = 4$.

- **Step 2:** $\Delta(B \mid \{A\}) = 1$ (adds $V_5$), $\Delta(C \mid \{A\}) = 1$ (adds $V_6$); break ties arbitrarily, say choose $B \Rightarrow S = \{A, B\}$ and $f = 5$.

- **Step 3:** $\Delta(C \mid \{A, B\}) = 1$; choose $C \Rightarrow f = 6$.

Thus greedy returns $\{A, B, C\}$ of size 3. With $f(G) = 6$, the harmonic bound is $H_6 \approx 2.45$, and $3 \leq 2.45 \times 2 = 4.9$ satisfies the guarantee.

# E. Additional Experiments

## E.1. Submodular Cover

Consider a universe of vulnerability types $U = \{V_1, \ldots, V_6\}$ and three available augmentations/patches $A, B, C$ that "cover" subsets of $U$:

$$A = \{V_1, V_2, V_3, V_4\}, \ B = \{V_1, V_2, V_5\}, \ C = \{V_3, V_4, V_6\}.$$

Let $f(S)$ be the number of unique vulnerabilities covered by $S \subseteq \{A, B, C\}$. The goal is to reach the target $|U| = 6$ with as few elements as possible. In this instance, the *optimal* solution uses two patches $\{B, C\}$, while the greedy algorithm selects $\{A, B, C\}$ (three patches) to reach the same target. As shown in Table 21, this matches the classical *submodular set cover* behavior where greedy is within a harmonic (logarithmic) factor of optimal.

*Table 21.* Toy coverage instance for submodular cover. Greedy attains the target with one more element than optimal (harmonic-factor behavior).

|  | Set | $|S|$ | $f(S)$ |
|---|---|---|---|
| Optimal | $\{B, C\}$ | 2 | 6 |
| Greedy | $\{A, B, C\}$ | 3 | 6 |

*Table 22.* Log-det surrogate on a 5-item ground set ($\sigma^2 = 0.5$). Greedy equals optimal for $k \in \{2, 3\}$ and exhibits diminishing returns (values rounded to 4 decimals).

|  | Selected $S$ | $g(S)$ |
|---|---|---|
| Greedy ($k$=2) | $\{g_1, g_3\}$ | 2.1972 |
| Optimal ($k$=2) | $\{g_1, g_3\}$ | 2.1972 |
| Greedy ($k$=3) | $\{g_1, g_3, g_5\}$ | 2.7000 |
| Optimal ($k$=3) | $\{g_1, g_3, g_5\}$ | 2.7000 |
| $\Delta\, g_5 \mid \{g_1\}$ | – | 0.8358 |
| $\Delta\, g_5 \mid \{g_1, g_3\}$ | – | 0.5028 |

# F. Experimental Protocol for Submodularity Verification

We empirically test whether the safety objective behaves approximately submodular when the "set elements" correspond to alignment/augmentation examples used for supervised fine-tuning (SFT). For any subset $S$, we define $f(S)$ as the fraction of adversarial test prompts for which the fine-tuned model produces a response judged **safe** by an external safety classifier/judge (e.g., Llama-Guard-3-8B), i.e., a safety score in $[0, 1]$. We set $f(\emptyset)$ to the base model's safety score without SFT.

To reduce confounds, we use a **deterministic evaluation mode**: (1) fixed random seeds across Python/NumPy/PyTorch/Transformers, (2) greedy (non-sampling) decoding during evaluation, and (3) a fixed training budget by disabling early stopping. Additionally, the number of training steps is scaled with $|S|$ so that each example is seen a comparable number of times across different subset sizes.

We perform two complementary checks. **Monotonicity** is tested by constructing nested subsets $S_1 \subset S_2 \subset \cdots$ and plotting $f(S_k)$ against $|S_k|$. **Diminishing returns** is tested by repeatedly sampling triples $(A, B, x)$ with $A \subset B$ and $x \notin B$, and comparing $\Delta(x \mid A)$ and $\Delta(x \mid B)$. We summarize results via (i) a scatter plot of $\Delta(x \mid A)$ vs. $\Delta(x \mid B)$ with the diagonal as the boundary, and (ii) the empirical violation rate $\mathbb{P}[\Delta(x \mid A) < \Delta(x \mid B)]$. Figure 1 illustrates these diagnostics.

## F.1. Log-Det Surrogate: Greedy vs. Optimal

We construct five candidate augmentations with two-dimensional features and use the PSD kernel $K_{ij} = \langle \phi(g_i), \phi(g_j) \rangle$ and a ridge parameter $\sigma^2 = 0.5$. For $S \subseteq G$, define the spectral objective $g(S) = \log \det(I + \sigma^{-2} K_{S,S})$. As illustrated in Table 22, for cardinalities $k = 2$ and $k = 3$, greedy *matches* the optimal value in this toy. We also observe diminishing returns for adding the same item in richer contexts, illustrating submodularity.

The toy demonstrates: (i) $g$ is normalized/monotone/submodular; (ii) greedy selection under a sized budget is $(1 - 1/e)$-approximate; (iii) marginal gains decrease as the context grows, consistent with diminishing returns.

## F.2. Cyclic Augmentation Reduces Sensitivity

We use a stylized multi-path model to visualize the mechanism behind the $1/n$ sensitivity reduction. Suppose a refusal is reached through $n$ equiprobable paths. An adversary degrades *one* specific path by an amount $\epsilon$. The overall refusal probability can be written as

$$H(\epsilon) \;=\; p(1 - \epsilon/n),$$

so the initial slope is $\frac{dH}{d\epsilon}\big|_{\epsilon=0} = -p/n$. Figure 2 plots $H(\epsilon)$ for $n \in \{1, 2, 4, 8\}$ with $p = 0.8$, showing the linear sensitivity shrinking exactly as $1/n$ (numerical slopes at $\epsilon$=0: $-0.8, -0.4, -0.2, -0.1$).

**Remark.** These compact toys show (i) why minimal augmentation is naturally a submodular cover; (ii) how the log-det surrogate yields a tractable, provably near-optimal selector; and (iii) why cyclic augmentation distributes refusal cues so that single-path perturbations are amortized by a factor of $n$.

## F.3. Real-World LLM Experiments

We validate our findings on instruction-tuned LLMs using QLoRA fine-tuning. Robustness is measured as $\mathcal{R}_\varepsilon = 1 - \text{ASR}$ on adversarial prompt sets. Unless noted, we report the mean over 3 seeds with 95% CIs.

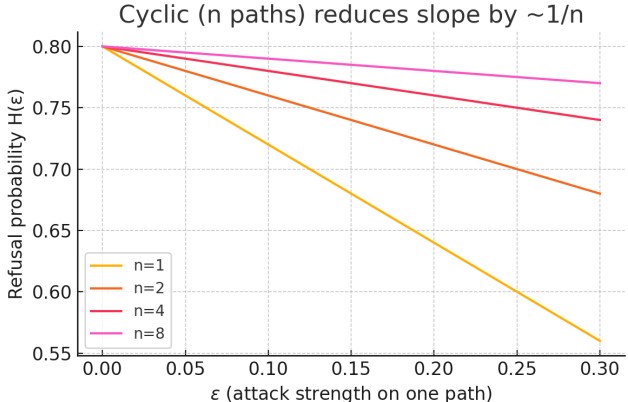

*Figure 2.* Stylized cyclic augmentation: $n$ parallel refusal paths with a single-path degradation of size $\epsilon$ gives $H(\epsilon) = p(1 - \epsilon/n)$; the initial slope scales as $-p/n$. Here $p = 0.8$, $n \in \{1, 2, 4, 8\}$.

**Experimental Setup.** We utilize three open-source instruction-tuned models: `Llama-3.1-8B-Instruct`, `Mistral-7B-Instruct`, and `Qwen-2.5-7B-Instruct`. Alignment training is performed using the UltraFeedback dataset (Cui et al., 2023). For the inverse-alignment perturbation phase, we use harmful prompts sourced from the AdvBench Harmful Behaviors subset(Zou et al., 2023a). All fine-tuning utilizes 4-bit QLoRA with the following hyperparameters: Rank $r = 16$, Alpha $\alpha = 32$, Dropout $p = 0.1$. We use the AdamW optimizer and a Cosine learning rate schedule with peak LR 2e-4.

**Cyclic Augmentation** $1/n$ **Sensitivity Scaling** We investigate the hypothesis (Theorem 6) that cyclic augmentation of order $n$ reduces the model's sensitivity ($|s_n|$) to inverse-alignment perturbations by a factor of $1/n$. We measure the sensitivity after training with $n \in \{1, 2, 4, 8\}$.

*Table 23.* Cyclic sensitivity scaling. Reported are $|s_n|$, ratios $\rho_n = |s_n|/|s_1|$, theory $1/n$, and initial $R_0$ with 95% CI width.

| Model | n | $|s_n|$ | $\rho_n$ | Theory $1/n$ | $R_0$ | 95% CI ($\pm$) |
|---|---|---|---|---|---|---|
| `Llama-3.1-8B-Instruct` | 1 | 4.990e-03 | 1.000 | 1.000 | 0.8598 | 4.23e-05 |
| | 2 | 3.090e-03 | 0.619 | 0.500 | 0.8731 | 3.72e-05 |
| | 4 | 2.558e-03 | 0.513 | 0.250 | 0.8718 | 3.74e-05 |
| | 8 | 2.403e-03 | 0.482 | 0.125 | 0.8804 | 3.15e-05 |
| `Mistral-7B-Instruct` | 1 | 3.213e-03 | 1.000 | 1.000 | 0.8355 | 3.62e-05 |
| | 2 | 9.259e-04 | 0.288 | 0.500 | 0.7874 | 2.98e-05 |
| | 4 | 1.441e-03 | 0.448 | 0.250 | 0.8748 | 4.39e-05 |
| | 8 | 1.076e-03 | 0.335 | 0.125 | 0.8846 | 3.86e-05 |
| `Qwen-2.5-7B-Instruct` | 1 | 2.787e-03 | 1.000 | 1.000 | 0.8349 | 4.34e-05 |
| | 2 | 2.671e-03 | 0.958 | 0.500 | 0.9011 | 3.50e-05 |
| | 4 | 6.611e-04 | 0.237 | 0.250 | 0.8450 | 4.20e-05 |
| | 8 | 1.942e-03 | 0.697 | 0.125 | 0.8584 | 3.00e-05 |

**Analysis of Scaling Deviations.** While Theorem 6.1 predicts an ideal $1/n$ reduction in sensitivity, Table 23 reveals significant deviations in practice. For instance, Llama-3.1-8B exhibits $\rho_8 = 0.482$, nearly four times the theoretical 0.125. This discrepancy suggests that the idealized assumptions of the theoretical model may not fully hold. Potential factors include the highly non-linear nature of deep neural networks, where information redundancy might be higher than predicted, or limitations imposed by parameter-efficient fine-tuning (QLoRA), which might restrict the model's capacity to effectively utilize the diversity introduced by augmentation.

### F.3.1. MINIMAL COVER TO REACH TARGET ROBUSTNESS

We evaluate the efficiency of the Greedy Cover algorithm for the minimal augmentation problem (Theorem 1). We set a target robustness $\mathcal{R}_{\text{target}}$ for each model and compare the size of the set $|S|$ required by Greedy Cover, the Optimal solution (found via ILP on the proxy objective), and a Random selection baseline.

*Table 24.* Minimal cover size to reach the target robustness (Greedy vs. Optimal Proxy vs. Random) with harmonic lower bound (LB).

| Model | Method | $|S|$ | Opt. Ratio | LB |
|---|---|---|---|---|
| Llama-3.1-8B-Instruct | Greedy | 2 | 1.00 | 2 |
| | Optimal (Proxy) | 2 | 1.00 | 2 |
| | Random | 2 | 1.00 | 2 |
| Mistral-7B-Instruct | Greedy | 2 | 1.00 | 2 |
| | Optimal (Proxy) | 2 | 1.00 | 2 |
| | Random | 5 | 2.50 | 2 |
| Qwen-2.5-7B-Instruct | Greedy | 2 | 1.00 | 2 |
| | Optimal (Proxy) | 2 | 1.00 | 2 |
| | Random | 5 | 2.50 | 2 |

**Analysis of Minimal Cover.**    Table 24 demonstrates the effectiveness of the submodular optimization approach. In all cases, the Greedy algorithm finds a solution identical in size to the Optimal proxy solution ($|S| = 2$), achieving the theoretical lower bound (LB). This significantly outperforms the Random baseline, which requires 2.5 times more augmentations for Mistral and Qwen, validating the necessity of principled selection.

### F.3.2. SAFETY–UTILITY TRADE-OFF

A critical concern with safety augmentations is the potential degradation of general utility. We measure the impact of our greedy augmentation selection on utility using a set of neutral prompts (e.g., MMLU subset (Hendrycks et al., 2021a;b)) independent of the safety augmentation.

*Table 25.* Safety–utility trade-off. $\mathcal{R}_\varepsilon$ and $\Delta$Util across selections.

| Model | Selection | $R_\epsilon(\uparrow)$ | $\Delta$Util$(\uparrow)$ |
|---|---|---|---|
| Llama-3.1-8B-Instruct | Baseline (S=∅) | 0.9179 | 0 |
| Llama-3.1-8B-Instruct | Greedy-n=2 | 0.8101 | +0.0296 |
| Llama-3.1-8B-Instruct | Greedy-n=4 | 0.9365 | -0.0079 |
| Llama-3.1-8B-Instruct | Greedy-n=8 | 0.9379 | -0.0205 |
| Mistral-7B-Instruct | Baseline (S=∅) | 0.8750 | 0 |
| Mistral-7B-Instruct | Greedy-n=2 | 0.8026 | -0.0453 |
| Mistral-7B-Instruct | Greedy-n=4 | 0.9721 | -0.0969 |
| Mistral-7B-Instruct | Greedy-n=8 | 0.9991 | -0.0546 |
| Qwen-2.5-7B-Instruct | Baseline (S=∅) | 0.8928 | 0 |
| Qwen-2.5-7B-Instruct | Greedy-n=2 | 0.8892 | +0.0505 |
| Qwen-2.5-7B-Instruct | Greedy-n=4 | 0.9378 | +0.0297 |
| Qwen-2.5-7B-Instruct | Greedy-n=8 | 0.9475 | -0.0001 |

**Analysis of Trade-offs.**    Table 25 presents the safety ($\mathcal{R}_\varepsilon$) and utility changes ($\Delta$Util). Excessive data augmentation can degrade model utility, while insufficient augmentation fails to provide adequate safety guarantees. To address this trade-off, we adopt a minimal augmentation strategy that achieves the desired safety threshold while minimizing the impact on downstream task performance. We observe that the impact on utility is generally small (within $\pm 5\%$, except Mistral $n = 4$). Notably, Qwen-2.5 shows simultaneous improvement in both safety and utility for $n = 2$ and $n = 4$. This suggests that minimal augmentation selection can enhance safety without significantly compromising general capabilities. We note an anomaly where Greedy-n=2 resulted in lower robustness for Llama and Mistral compared to the baseline, which warrants further investigation into the interaction dynamics and optimization stability of small augmentation sets.

**Diminishing Returns, $\widehat{\gamma}_k$, and $\widehat{c}$**    We empirically estimate the weak-submodularity ratio $\widehat{\gamma}_k$ and curvature $\widehat{c}$ of the robustness function. This validates the diminishing returns property and informs the theoretical guarantees of the Greedy algorithm (Theorems 4 and 5). We fix an augmentation $a$ and evaluate its marginal gain $\Delta(a \mid S)$ over various contexts $|S|$.

*Table 26.* Estimated weak-submodularity ratio $\widehat{\gamma}_k$ and curvature $\widehat{c}$.

| Model | $k$ | $\widehat{\gamma}_k$ | $\widehat{c}$ |
|---|---|---|---|
| Llama-3.1-8B-Instruct | 4 | 0.9959 | 1.0000 |
| Mistral-7B-Instruct | 4 | 0.9983 | 1.0000 |
| Qwen-2.5-7B-Instruct | 4 | 0.9952 | 1.0000 |

**Interpretation of Submodularity Parameters.**    The results in Table 26 are intriguing. The estimated weak-submodularity ratio $\widehat{\gamma}_k$ is very close to 1 (e.g., 0.9959 for Llama-3.1). This suggests that the robustness function is nearly modular (additive) over the tested augmentations ($k = 4$), implying minimal diminishing returns in this regime. Conversely, the estimated curvature $\widehat{c}$ is 1.0, indicating maximum curvature (standard submodularity, where marginal gains can drop significantly depending on the context). The coexistence of near-modularity and high curvature implies a specific structure: while the augmentations provide nearly independent value across most contexts (supporting the high $\gamma_k$), there exists at least one context where adding a specific augmentation provides negligible additional benefit (supporting $c = 1$). This complex landscape confirms that submodular optimization is appropriate, even if additive effects dominate locally.

