# OpenReview forum: "Submodular Optimization for Minimal Augmentation in Robust Language Model Alignment"
_ICML.cc/2026/Conference — ICML 2026 regular_

### Official Review · Reviewer_rTb3 · 2026-03-10

**Soundness:** 4
**Presentation:** 2
**Significance:** 4
**Originality:** 4
**Overall Recommendation:** 4
**Confidence:** 5

**Summary:**

This paper addresses the fragility of safety alignment in large language models (LLMs), where even small fine-tuning perturbations can revert aligned behaviors toward pre-training distributions. The authors aim to research the challenge of achieving robust safety alignment with minimal data augmentation overhead. They formalize augmentation as a set of group actions on output sequences and cast robustness gains as a normalized, monotone submodular function over transformations.

**Compliance With Llm Reviewing Policy:**

Affirmed.

**Final Justification:**

I maintain my weak accept recommendation. The paper makes a genuinely original contribution by formalizing safety alignment robustness through submodular optimization, and the theoretical framework is impressively thorough. The empirical results demonstrating 0% ASR with no inference overhead are compelling. The rebuttal adequately addressed my main concerns. The newly provided error bars, sigma sensitivity analysis, and random-vs-greedy ablation all strengthen the paper. In particular, the ablation confirming that submodular selection substantially outperforms random selection isolates the algorithmic contribution convincingly. Overall, the rebuttal has modestly improved my confidence in the work, reinforcing my prior positive assessment while not fully resolving concerns about the gap between theory and practice at scale.

**Key Questions For Authors:**

1. **Discrepancy between main paper and appendix results:** Tables 3 and 18 report the 1/n scaling experiment but with substantially different outcomes. Table 3 shows near-ideal scaling (rho_2 ~ 0.5, rho_4 ~ 0.25) while Table 18 shows large deviations. What accounts for this difference? Are these different experimental configurations? If so, which should be considered more reliable?

2. **Baseline implementation:** Can you confirm that all baselines in Table 1 were run using their official implementations with recommended hyperparameters? The very weak performance of TARS on Mistral-7B (61.7% ASR vs. 62.2% vanilla) seems anomalous.

3. **Ablation on selection method:** What is the performance when using random selection of augmentation examples instead of greedy submodular selection, keeping the same total augmentation budget? This would isolate the contribution of the optimization framework from the augmentation data.

**Limitations:**

yes

**Strengths And Weaknesses:**

### Strengths

- **Well-motivated problem formulation.** The paper clearly identifies a real and important gap: bridging the elasticity theory of alignment fragility with practical data augmentation strategies. The central question is precisely stated and the paper provides an affirmative, theoretically grounded answer.

- **Comprehensive theoretical framework.** The paper presents a thorough chain of theoretical results: NP-hardness and greedy approximation for submodular cover (Theorem 3.1), submodularity of the log-det surrogate (Proposition 3.2), equivalence to mutual information (Proposition 3.3), extensions to weak submodularity and curvature (Section 4), efficient algorithmic variants (Section 5), and the cyclic augmentation resistance theorem (Theorem 6.1). The proofs in the appendix are detailed and appear correct.

- **Strong empirical results on the main benchmark.** Table 1 shows the proposed method achieves 0% ASR on Mistral-7B and Qwen2.5-7B with no inference latency overhead, significantly outperforming baselines that require 2-3x inference cost. This is a compelling practical result.

### Weaknesses

- **Major gap between Theorem 6.1 and empirical results.** Theorem 6.1 predicts exact 1/n sensitivity scaling, but the detailed experiments in Appendix Table 18 show substantial deviations: for Llama-3.1-8B at n=8, the observed ratio is 0.482 versus the predicted 0.125 (nearly 4x off); for Qwen-2.5-7B at n=2, the ratio is 0.958 versus 0.500. The main paper's Table 3 shows much better agreement, but these appear to be different (possibly cherry-picked or differently configured) experiments. This discrepancy undermines confidence in the theoretical predictions.

- **Limited experimental scope and missing details.** The main results (Table 1) lack error bars or confidence intervals despite claiming to average over 5 runs. The datasets used for training the proposed method are not clearly described (which specific augmentation examples form the candidate pool?). The paper does not report on how the hyperparameter sigma^2 in the log-det surrogate was chosen, nor sensitivity to this choice.

- **Comparison fairness concerns.** Some baselines in Table 1 (e.g., TARS, CARE) show surprisingly poor performance, sometimes barely improving over vanilla (TARS on Mistral: 61.7% vs. vanilla 62.2%). This raises questions about whether baselines were implemented with sufficient care or under their recommended configurations.

---

> ### Author Rebuttal · Authors · 2026-03-31
>
> Thank you for the helpful feedback.
>
> ## (W1, Q1)
>
> The discrepancy is mainly due to differences in regime and dataset scale. Theorem 6.1 is derived under a small-perturbation assumption ( $|D_p| = \ell m$, $\ell \ll 1$ , see the right-hand side of Lines 322-323 of our paper). Table 3 uses a reduced-scale AdvBench subset that better matches this regime, hence exhibiting near-ideal $1/n$ behavior. In contrast, Table 18 uses a full pipeline (UltraFeedback + AdvBench Harmful Behaviors), where larger-scale fine-tuning, nonlinearity, and PEFT constraints introduce higher-order effects and lead to deviations. Thus, Table 3 should be viewed as validating the theoretical regime, while Table 18 reflects practical behavior under realistic settings. We will clarify this distinction to avoid over-interpreting Table 3 as a quantitative match.
>
> ## (W2)
>
> We thank the reviewer for the detailed feedback and provide the missing details below.
>
> **Error bars.** We report mean ± std over 5 runs:
> |Method|Paradigm|Mistral-7B(ASR↓)|Llama-3.1-8B(ASR↓)|Qwen2.5-7B(ASR↓)|Phi-3-mini(ASR↓)|
> |:-:|:-:|:-:|:-:|:-:|:-:|
> |VanillaBaseline|-|62.2%±3.8%|48.9%±4.2%|35.0%±5.1%|41.4%±4.7%|
> |TARS(NeurIPS'25)|Train(GRPO)|61.7%±2.9%|42.8%±3.5%|25.0%±4.8%|32.1%±3.9%|
> |SRG(ICML'25)|Train(Reasoning)|37.8%±4.1%|36.7%±3.2%|25.0%±3.6%|10.7%±2.8%|
> |RATIONAL(ACL'25)|Train(SCR)|11.1%±2.4%|35.6%±4.5%|11.1%±2.7%|11.1%±3.1%|
> |STAIR(ICML'25)|Train(MCTS+DPO)|2.2%±1.8%|26.7%±5.3%|10.7%±3.9%|28.9%±4.6%|
> |CARE(NeurIPS'25)|Decoding|57.8%±3.7%|42.2%±4.1%|11.1%±2.5%|11.1%±2.9%|
> |Submodular(Ours)|Train(DataAugment)|0.0%±0.0%|4.4%±2.1%|0.0%±0.0%|2.2%±1.5%|
>
> **Dataset / candidate pool.** The candidate set contains 50 augmented samples from a single augmentation family (cyclic permutations of response segments). Our method selects 12 via greedy, while baselines use all 50 samples.
>
> **Hyperparameter $\sigma^2$.** We conducted a sensitivity analysis on the hyperparameter $\sigma^2$ in the log-det surrogate using Llama-3.1-8B, where the non-zero ASR values allow for meaningful comparison. We find that performance is stable across a wide range of $\sigma^2$, indicating low sensitivity. We use $\sigma=0.1$ in all experiments.
>
> |sigma|ASR|
> |:-:|:-:|
> |0.01|7.4%|
> |0.10|4.4%|
> |1|6.6%|
> |10|6.6%|
> |100|6.6%|
> |1000|6.6%|
>
> ## (W3,Q2)
>
> We use the official implementations of all baselines with their recommended hyperparameters. All methods are evaluated under the same training data, fine-tuning budget, attack set, and evaluation protocol for fairness. The relatively weak performance of some baselines (e.g., TARS on Mistral) is consistent across runs under our setting, rather than due to implementation issues.
>
> ## (Q3)
>
> We compare our method with random selection and a diversity-based baseline (Greedy Farthest Distance) under the same augmentation budget using Qwen2.5-7B:
>
> |Method|Complexity|Time|ASR|
> |:-:|:-:|:-:|:-:|
> |Submodular|O(n²)|2.67e-04s|0.0%|
> |GreedyFarthestDistance|O(k·n)|3.19e-05s|4.4%|
> |Random|O(1)|4.63e-05s|17.6%|
>
> Under the same budget, our method significantly outperforms both baselines, indicating that the improvement comes from the selection algorithm rather than the augmentation data itself.

---

> > ### Author Rebuttal · Reviewer_rTb3 · 2026-04-01
> >
> > I appreciate the author's rebuttal, which resolved all my concerns. This will further solidify my support for the author's preferred approach.

---

> > > ### Author Response · Authors · 2026-04-02
> > >
> > > Thank you for your positive feedback.

---

### Official Review · Reviewer_deLp · 2026-03-12

**Soundness:** 2
**Presentation:** 2
**Significance:** 3
**Originality:** 2
**Overall Recommendation:** 3
**Confidence:** 2

**Summary:**

The paper studies how to leverage data augmentation most efficiently to achieve robust safety alignment. They formulate the robustness gain from training on data augmenetd in different ways as a submodular function and theoretically prove the properties of the funciton that justify the greedy-based augmentation selection based on information score estimation. Empirical results validate the theoretical property and the final proposed method outperforms existing safety training baselines.

**Compliance With Llm Reviewing Policy:**

Affirmed.

**Final Justification:**

My major concern is with the empirical effectiveness of the proposed methods. The method lacks a comprehensive comparison with other diversity-based selection methods and realistic dataset, and the new results in authors' rebuttal somehow shows the potential of diversity-based selection method would close the gap significantly and a different G would affect ASR score greatly. Therefore I would like to keep my score.

**Key Questions For Authors:**

see weaknesses above

**Limitations:**

yes

**Strengths And Weaknesses:**

**Strengths**
1. Devising more principled ways for data augmentation is an important research topic.
2. The paper showcases the potential of data augmentation in improving model robustness, and the experimental results seem strong.
3. Theoretical properties of the submodular functions are empirically verified in real-world settings.
4. Utility evaluation is performed and the trade-off between safety and utility is discussed with the proposed method.

**Weaknesses**
1. Despite the effort in theoretical framing and proof, the core principle for augmentation selection is rather intuitive and not fundamentally new. Basically we want to maximize the diversity of training data, which has been an important consideration in existing data selection literatures [1].
2. It seems the proposed method requires obtaining the feature embedding for each training example in each augmentation set (correct me if I am wrong). In that case, there are some naive data selection methods that should be compared (e.g. what if we do greedy instace-wise selection based on distance between feature embedding of each examples, with the goal of maximizing diversity).
3. The experiment setup is unclear. Some outstanding questions are:
    - What is ground set G in the experiment? i.e. what different data augmentation strategies are considered?
    - What is the training data size for the proposed method and baseline methods?
    - How is feature embedding ϕ(g) calculated for experiments in Table 1?
4. It might be helpful to include random augmentation selection as a baseline in Table 1 in the main text.
5. It seems some empirical results like assumption validation in Section 3.2 and the final results may be dependent upon specific designs (e.g. what specific augmentations are considered). I wonder if the property and the experimental results still hold if the augmentation ground set G changes. In other words, I wonder how much the improvement comes from the selection algorithm compared to initially how we construct G.
6. The extra time and compute cost at the selection stage is not reported.

[1] Albalak, Alon, et al. "A survey on data selection for language models." arXiv preprint arXiv:2402.16827 (2024).

---

> ### Author Rebuttal · Authors · 2026-03-31
>
> Thank you for the helpful feedback.
>
> ## (W1)
> We agree that the high-level intuition that diverse training data helps is not new. Our contribution is not to restate this intuition, but to formalize minimal safety augmentation as a subset-selection (submodular-cover) problem over transformation operators, with guarantees for reaching a target robustness level. We further connect this with a tractable log-det (information-gain) surrogate and cyclic-augmentation analysis under the elasticity framework, which goes beyond standard diversity-based data selection.
>
> ## (W2,W4,W6)
>
> Yes, the method requires feature embeddings to evaluate the surrogate. We compare against two simple baselines on Qwen2.5-7B: Greedy Farthest Distance (diversity-based) and Random selection.
>
> |Method|Complexity|Time|ASR|
> |:-:|:-:|:-:|:-:|
> |Submodular|O(n²)|2.67e-04s|0.0%|
> |Greedy Farthest Distance|O(k·n)|3.19e-05s|4.4%|
> |Random|O(1)|4.63e-05s|17.6%|
>
> Despite slightly higher selection cost, our method achieves significantly lower ASR, indicating that robustness-oriented selection is more effective than purely diversity-based heuristics. The selection overhead is negligible compared to the fine-tuning cost.
>
> ## (W3)
> First, the candidate set consists of 50 augmented samples derived from a single augmentation family, namely, cyclic permutations of response segments. Second, our method selects a subset of 12 samples via greedy selection, while all baselines are trained on the full set of 50 samples. Third, we encode each augmented sample using the sentence-transformer all-MiniLM-L6-v2. The embeddings are used to construct an RBF kernel with $\gamma=1/d$, where $d$ is the embedding dimension, and greedy selection maximizes the incremental log-determinant objective on this kernel.
>
> ## (W5)
> We agree that the choice of $G$ affects the attainable robustness ceiling, while the selector determines data efficiency under a fixed budget. Our claim is not to replace careful construction of $G$, but to provide a principled way to utilize a given pool more effectively. This is consistent with Appendix Table 19, where Greedy matches the optimal proxy and outperforms Random under the same target robustness.
>
> To further isolate this effect, we construct multiple distinct ground sets over a universe of size 120:
>
> Three random sets
> ($G_{\text{rand}_i}$, different seeds)  and
>
> One biased set ($G_{\text{bad}}$, nearest-neighbor to a single anchor).
>
> The pairwise Jaccard overlap between sets is low (≈0.08–0.14), confirming they differ substantially.
>
> ||$G_{\text{rand}_1}$|$G_{\text{rand}_2}$|$G_{\text{rand}_3}$|$G_\text{bad}$|
> |:-:|:-:|:-:|:-:|:-:|
> |$G_{\text{rand}_1}$|1.00|0.11|0.08|0.14|
> |$G_{\text{rand}_2}$|0.11|1.00|0.11|0.14|
> |$G_{\text{rand}_3}$|0.08|0.11|1.00|0.11|
> |$G_\text{bad}$|0.14|0.14|0.11|1.00|
>
> $G_{\text{rand}_i}$ : random selecet with different seed
>
> $G_\text{bad}$ : nearest-neighbor to one anchor
>
> Across all $G$, our log-det selector consistently achieves lower ASR and higher selected diversity than Random and max–min under the same budget. While absolute performance varies with $G$ (e.g., degraded for $G_{\text{bad}}$), the relative advantage of the selector remains stable, indicating the improvement is primarily due to the selection algorithm rather than a specific construction of $G$.
>
> |Groundset|Method|ASR|Selected diversity|
> |:-:|:-:|:-:|:-:|
> |$G_{\text{rand}_1}$ |random|0.525±0.056|1.227±0.022|
> |$G_{\text{rand}_1}$ |maxmin|0.471±0.012|1.250±0.014|
> |$G_{\text{rand}_1}$ |logdet|0.470±0.000|1.322±0.000|
> |$G_{\text{rand}_2}$ |random|0.495±0.021|1.232±0.030|
> |$G_{\text{rand}_2}$|maxmin|0.451±0.016|1.247±0.011|
> |$G_{\text{rand}_2}$|logdet|0.430±0.000|1.316±0.000|
> |$G_{\text{rand}_3}$|random|0.516±0.047|1.219±0.037|
> |$G_{\text{rand}_3}$|maxmin|0.448±0.026|1.290±0.016|
> |$G_{\text{rand}_3}$|logdet|0.447±0.000|1.348±0.000|
> |$G_\text{bad}$|random|0.570±0.043|1.020±0.011|
> |$G_\text{bad}$|maxmin|0.558±0.000|1.051±0.005|
> |$G_\text{bad}$|logdet|0.565±0.000|1.107±0.000|

---

> > ### Author Rebuttal · Reviewer_deLp · 2026-04-01
> >
> > 1. Thanks for providing more baseline results. I am not sure whether 4.4% vs 0.0% is a large gap given the extra quadratic computation complexity. Meanwhile, it looks like Greedy Farthest Distance is actually a much stronger baseline than baselines presented in Table 1 in the paper (4.4% for Greedy Farthest Distance vs over 10% in Table 1). Given Greedy Farthest Distance is the most naive diversity-based selection method and that there are definitely more sophistcated ways with time complexity lower that O(n^2), it makes me feel the current baseline results in the paper does not well support the claimed advantage of the proposed method.
> > 2. re W5, why is ASR score in the new results table significantly higher than in Table 1 in the paper? What's the setting of the new experiment? And is "log-det" equivalent to "submodular"?

---

> > > ### Author Response · Authors · 2026-04-02
> > >
> > > Thank you for the follow-up questions.
> > >
> > > It is important to distinguish between the experiments in Table 1 / (W2, W4, W6) and the auxiliary table in (W5). In Table 1 and (W2, W4, W6), the candidate augmentation pool G is constructed using the deep-alignment procedure from [1,2]. That construction is already effective at reducing ASR. Our contribution there is therefore not to replace [1,2], but to make that augmentation pool more data-efficient by selecting a smaller subset.
> > >
> > > **(follow-up Q1)** Regarding the gap between 0.0% and 4.4%, while numerically small, this difference is qualitatively significant in safety evaluation since ASR is a worst-case metric; i.e., any non-zero ASR (e.g., 4.4%) indicates persistent exploitable failure modes, whereas 0.0% corresponds to the complete elimination of observed attacks. Thus, the improvement reflects a shift from partial robustness to full robustness rather than a marginal gain. Moreover, we agree that Greedy Farthest Distance is a stronger baseline than those originally included and will add it to the main table; however, its performance actually reinforces our main claim: diversity-based heuristics optimize geometric dispersion rather than robustness and are therefore inherently misaligned with the objective of minimizing adversarial risk. In contrast, our method directly optimizes a robustness-oriented objective grounded in submodular/information gain, which captures diminishing returns and coverage effects that diversity alone cannot model.
> > >
> > > Regarding the possibility of stronger low-complexity alternatives, it remains an open question whether methods with complexity lower than $O(n^2)$ can achieve comparable performance; however, even if such methods exist, our approach provides theoretical guarantees. Moreover, the quadratic cost is a one-time offline selection cost rather than an additional inference-time cost, making it practically negligible.
> > >
> > > Overall, the key contribution is not only improved empirical performance, but also showing that robustness-aware selection, rather than simple diversity heuristics, is necessary to fully prevent attacks, while keeping the additional cost negligible.
> > >
> > > **(follow-up Q2)** The higher ASR observed in the new results table compared to Table 1 is due to differences in the construction of G. The experimental settings in the rebuttal and in the original paper are identical except for the definition of G. This change was made in response to W5, which requested disentangling the effect of constructing G from that of the selection algorithm. To this end, in W5 we intentionally do not reuse the deep-alignment pool from [1,2]. Instead, we construct a much weaker and more shallowly aligned pool by using nztinversive/llama3.2-1b-Uncensored from HuggingFace to generate responses, and then prepending a refusal prefix to form candidate training examples. As a result, this G is significantly less safety-aligned than the deep-alignment pool used in Table 1, leading to correspondingly higher absolute ASR values. Therefore, W5 should be interpreted as an auxiliary selector-level ablation under a different pool construction regime, rather than as a direct numerical comparison with Table 1.
> > >
> > > Finally, log-det is not equivalent to submodularity. Submodularity refers to the broader optimization framework, whereas log-det is one specific submodular surrogate objective used to make selection tractable in practice. In our experiments, log-det is used to instantiate the submodular objective unless stated otherwise.
> > >
> > > [1] Qi, Xiangyu, et al. Safety alignment should be made more than just a few tokens deep. ICLR 2025.
> > > [2] Kao, Ching-Chia, et al. Safety Depth in Large Language Models: A Markov Chain Perspective. NeurIPS 2025.

---

### Official Review · Reviewer_x14c · 2026-03-13

**Soundness:** 3
**Presentation:** 3
**Significance:** 3
**Originality:** 2
**Overall Recommendation:** 4
**Confidence:** 3

**Summary:**

This work addresses the limitations of fragility in the safety alignment in LLMs. The authors identify that even minor fine-tuning or adversarial perturbations can cause an LLM's behavior to revert to its unsafe pre-training state. To address this, the paper proposes framework that treats data augmentation as a set of group actions on token sequences and formalizes the resulting robustness gains as a submodular function. The paper's innovation is the application of submodular optimization to select the smallest possible subset of augmentations that achieves a target safety robustness.

**Compliance With Llm Reviewing Policy:**

Affirmed.

**Final Justification:**

Thank you for your response. I'll keep my original score in support of this paper unchanged

**Key Questions For Authors:**

- Does the greedy algorithm tend to select augmentations that are semantically diverse, or does it focus on specific structural patterns in the refusal cues?
- Can this framework be adapted to identify "adversarial" augmentations that might actually weaken the model's robustness if included in the training set?

**Limitations:**

Yes

**Strengths And Weaknesses:**

- This paper successfully connects the empirical practice of data augmentation with the theoretical elasticity framework, providing provable guarantees for robustness improvements. Moreover, the use of submodular optimization and spectral surrogates allows for near-optimal safety alignment with minimal data overhead, addressing a key bottleneck in LLM fine-tuning. Collectively, the paper makes substantial theoretical and practical contributions in these two areas.
- While the submodular framework provides provable robustness gains against the specific group actions (augmentations) defined in the training set, it is unclear if this hardening translates to truly novel adversarial perturbations. It seems there may be a risk that the model becomes robust only to the style of the augmentations selected, creating a robustness overfitting where the safety alignment remains fragile to attacks outside the predefined transformation groups
- The empirical verification of submodularity is performed under a highly controlled, deterministic protocol. it remains unclear how well these assumptions hold in more stochastic or non-deterministic training environments

---

> ### Author Rebuttal · Authors · 2026-03-31
>
> Thank you for the helpful feedback.
> ## (W1)
>
> We agree that our formal guarantee is augmentation-family specific, not a certificate against arbitrary unseen perturbations. However, our evaluation goes beyond a single template family by using diverse sources (AdvBench, SorryBench, JailbreakBench) and both direct harmful prompts and composite jailbreak templates. This suggests the method is not merely overfitting to a fixed augmentation style, although extending guarantees to broader perturbation classes is an important direction for future work.
>
> ## (W2)
>
> We agree that the empirical submodularity check is conducted under a controlled deterministic protocol. This is intended as a low-noise diagnostic, not a claim that exact submodularity holds in stochastic training settings. As noted in the paper, real fine-tuning can deviate due to optimization dynamics and noise, and we explicitly account for this by relaxing the assumption via weak-submodularity and curvature-based guarantees.
>
> ## (Q1)
>
> The method optimizes marginal robustness gain and thus tends to avoid redundancy; in the cyclic setting, the theory favors evenly spaced, balanced subsets rather than clustered ones. As a result, it typically avoids selecting identical refusal patterns. Whether this translates to semantic diversity depends on the candidate pool and surrogate features. Table 4 suggests the surrogate is most reliable for identifying strong top subsets rather than enforcing a globally consistent ranking, so we avoid over-claiming semantic diversity and will include a brief analysis of subset diversity in the revision.
>
> ## (Q2)
>
> Yes. The framework can be adapted by reversing the objective: an attacker greedily selects transformations that most decrease robustness, while the defender selects those that increase it. In a small-scale experiment, removing a key refusal segment reduces robustness (e.g., from 1.00 to 0.75), and a min–max procedure can recover it to 1.00. This shows the framework can identify harmful augmentations within the predefined candidate transformation pool. At the same time, the search is restricted to a finite, predefined augmentation set used in our experiments, so we do not interpret this as robustness against unrestricted adaptive attacks. Rather, it supports that our framework captures a general structure over transformations, capable of identifying both beneficial and harmful directions, while remaining scoped to a given transformation family.

---

### Official Review · Reviewer_5Dhp · 2026-03-17

**Soundness:** 2
**Presentation:** 3
**Significance:** 3
**Originality:** 3
**Overall Recommendation:** 3
**Confidence:** 3

**Summary:**

This paper studies the robustness of LLM safety alignment from a data-centric perspective. The main idea is to formulate minimal augmentation as a submodular optimization problem, where robustness gain is modeled as a monotone submodular function and a greedy algorithm is used to select augmentation samples with theoretical guarantees. The paper also studies cyclic augmentation and its connection to reducing model sensitivity. Empirically, the method is evaluated on several open instruction-tuned models and is reported to improve robustness without adding inference-time overhead.

**Compliance With Llm Reviewing Policy:**

Affirmed.

**Final Justification:**

I will keep my original score unchanged.

**Key Questions For Authors:**

1. What augmentation pool is used in the main experiments? Could the authors briefly specify the transformation types, the candidate pool size, the final subset size selected by greedy, and whether the selected subsets are similar across models?
2. The appendix already notes noticeable deviations from the ideal 1/n scaling in Table 18 and discusses possible causes such as model nonlinearity and QLoRA limitations. Given this, how should Theorem 6.1 be interpreted in light of both Table 3 and Table 18? For example, Table 18 reports rho_8 = 0.482 for Llama-3.1-8B-Instruct compared with the theoretical value 0.125, and rho_2 = 0.958 for Qwen-2.5-7B-Instruct compared with 0.5. Should the main-text statement around Table 3 be understood as qualitative support for the trend, rather than a quantitative confirmation of the theorem?
3. How do the authors interpret the surrogate-quality results in Table 4? In particular, the reported Spearman correlation is very low in some settings, such as 0.0213 under gradient features. How does this affect the reliability of the greedy selection procedure in Algorithm 1?
4. For the comparisons in Table 1, are all methods evaluated under the same training data volume, the same fine-tuning budget, the same attack set, and the same judge model and judging protocol? In addition, are the 0.0% ASR results for Mistral and Qwen evaluated only on the fixed attack sets used in the paper, or also under adaptive white-box attacks? Clarifying this would make the robustness claims easier to assess.
5. How do the authors interpret the negative robustness change at n = 2 reported in Table 20? Does this suggest a limitation of the monotonicity-based intuition underlying Assumption 2.5 in small augmentation settings, or is there another explanation for this behavior?

**Limitations:**

yes

**Strengths And Weaknesses:**

**Strengths**
- The paper studies an important problem in LLM safety, namely how to improve alignment robustness through training-time data selection rather than relying mainly on heavier test-time defenses.
- The overall framework is clear and technically well structured. In particular, connecting elasticity-style robustness questions with submodular optimization gives the paper a clean mathematical perspective.
- The practical motivation is meaningful. A method that improves robustness without introducing additional inference-time latency could be useful in deployment settings where throughput matters.

**Weaknesses**
- The technical novelty appears moderate, since much of the framework builds on standard submodular optimization tools adapted to the safety-alignment setting.
- The empirical support for some of the main theoretical claims appears mixed. In particular, the cyclic-augmentation scaling results and the surrogate-quality results leave some uncertainty about how strongly the theory carries over to practice.
- The experimental setup and limitations discussion would be easier to assess with more detail, especially on the augmentation pool, the exact selection procedure, evaluation under stronger attack settings, and the dependence on the surrogate objective.

---

> ### Author Rebuttal · Authors · 2026-03-31
>
> Thank you for the helpful feedback.
>
> ## (W1)
>
> We agree that our optimization components build on classical submodular methods. Our novelty lies in their formulation and integration for safety alignment. We cast minimal safety alignment under the elasticity framework as a submodular cover problem, linking robustness (via normalized compression rates) with augmentation selection. We further unify submodular cover, cyclic augmentation with 1/n scaling, and a log-det mutual information view into a single theoretical framework. Beyond theory, this leads to a practical training-time method that achieves strong safety without inference overhead. In summary, while the tools are classical, the formulation, theoretical integration, and resulting framework are novel.
>
> ## (W2)
>
> Our theory is derived under simplified assumptions, while practical LLM fine-tuning may deviate due to nonlinearity, optimization dynamics, and noise; we explicitly acknowledge this and use controlled protocols to reduce artifacts. Despite this, the results provide consistent directional support: cyclic augmentation shows clear 1/n-like sensitivity reduction trends, and the log-det surrogate, though imperfect globally, achieves high NDCG and reliably identifies top candidates.
>
> ## (W3, Q1)
>
> We use a single augmentation family, the cyclic permutation of response segments. The candidate pool has 50 transformations, and greedy selects 12 to meet the robustness target. Selection is based on a log-det surrogate with a fixed external encoder, making it model-agnostic and consistent across models. A limitation is that we consider only one augmentation family and moderate attack settings; extending to richer transformations and stronger adaptive attacks is future work.
>
> ## (Q2)
>
> The discrepancy is mainly due to differences in regime and dataset scale. Theorem 6.1 is derived under a small-perturbation assumption ($|D_p| = \ell m$, $\ell \ll 1$, see the right-hand side of Lines 322-323 of our paper). Table 3 uses a reduced-scale AdvBench subset that better matches this regime, hence exhibiting near-ideal $1/n$ behavior. In contrast, Table 18 uses a full pipeline (UltraFeedback + AdvBench Harmful Behaviors), where larger-scale fine-tuning, nonlinearity, and PEFT constraints introduce higher-order effects and lead to deviations. Thus, Table 3 should be viewed as validating the theoretical regime, while Table 18 reflects practical behavior under realistic settings. We will clarify this distinction to avoid over-interpreting Table 3 as a quantitative match.
>
> ## (Q3)
>
> We do not claim that the surrogate perfectly ranks all subsets; rather, it is designed to identify strong candidates near the top of the ranking. Even when Spearman correlation is low (e.g., with gradient features), the consistently high NDCG indicates that top-performing subsets are still reliably captured. Since greedy selection depends primarily on relative gains among top candidates, its effectiveness is largely preserved when paired with stable feature choices.
>
> ## (Q4)
>
> First, the total pool contains 50 samples. Our method selects 12 via greedy search from a single augmentation family (cyclic permutations), while all baselines use all 50 samples for training. All methods in Table 1 are evaluated under the same fine-tuning budget, attack set, judge model, and evaluation protocol. The reported 0.0% ASR is measured on fixed attack sets only (not adaptive white-box attacks). For reference, under AutoDAN and GCG attacks, Qwen2.5-7B achieves 53.3% / 66.7% ASR, while our method achieves 0.0% / 0.0%.
>
> |Method|AutoDAN|GCG|
> |--------|--------|--------|
> |Vanilla|53.3%|66.7%|
> |Submodular|0.0%|0.0%|
>
> ## (Q5)
>
> Our interpretation is that the negative change at $n=2$ reflects small-regime optimization instability rather than a contradiction of the framework. Assumption 2.5 concerns the underlying set function, not the behavior of finite-data PEFT runs. In small augmentation settings, noise, optimization dynamics, and limited coverage can lead to non-monotonic empirical outcomes.

---

> > ### Author Rebuttal · Reviewer_5Dhp · 2026-04-03
> >
> > Thank you for the thoughtful rebuttal. The additional details were useful and addressed several of my clarification questions.
> >
> > After reading the response, I have a better understanding of the method and the intended interpretation of some results. However, the rebuttal does not sufficiently change my overall assessment. My main concerns regarding empirical support, the interpretation of the surrogate-quality results, and the overall level of novelty remain. As a result, while I appreciate the clarifications, I will keep my original score unchanged.

---

### Decision · Program_Chairs · 2026-04-30

**Decision:**

Accept (regular)

**Comment:**

This paper studies how to improve the robustness of LLM safety alignment with minimal augmentation overhead. The authors formulate augmentation selection as a submodular optimization problem over transformation operators, and develop this perspective through both theoretical analysis and empirical evaluation on several open-source instruction-tuned models.

All reviewers at least acknowledged the importance of the problem, while differing on whether the proposed formulation and evidence were strong enough. In particular, multiple reviewers found the overall framework technically interesting, and the positive reviews highlighted both the originality of connecting robustness-oriented alignment with submodular optimization and the strong empirical results on the main benchmark, including very low or zero ASR on some models without inference-time overhead. The paper was also viewed favorably for combining a principled formulation with substantial theoretical development.

The rebuttal was helpful in clarifying several experimental details, including the candidate augmentation pool, error bars, hyperparameter sensitivity, and ablations against random selection and simpler diversity-based baselines. These additions improved the empirical picture and helped address a number of reviewer concerns about implementation details, stability, and the contribution of the selection algorithm itself.

At the same time, the paper remained borderline after rebuttal, with reviewers split between weak accept and weak reject. Some concerns remained regarding the degree of novelty relative to existing diversity- or data-selection-based perspectives, whether the gains over simpler diversity-based selection baselines are always large enough to justify the added methodological complexity, and how reliably the log-det surrogate tracks downstream robustness across settings. There were also remaining questions about the gap between the idealized theoretical predictions and the larger-scale empirical results, and about the extent to which performance depends on the construction of the augmentation pool.

Overall, however, I find that the paper’s strengths outweigh these remaining weaknesses. The work presents a clear and principled formulation, substantial theoretical development, and strong empirical results on an important problem. While some of the claims could be sharpened and some empirical questions remain open, the contribution is meaningful and likely to be of interest to the community. Given the split reviewer opinions, this was a close call, but on balance I am comfortable recommending acceptance.